# Respondents with more extreme views show moderation of opinions in multi-year surveys in the USA and the Netherlands

Nadav Klein [1✉] & Olga Stavrova [2,3✉]

People with extreme political attitudes are often assumed to be more resistant to change than moderates. If this assumption is true, extreme attitudes would ossify and continuously aggravate intergroup conflict and polarization. To test this assumption of stubborn extremists, we use large-scale panel surveys of attitudes towards policy issues and general ideologies across up to 13 years (combined $N = 16,238$). By tracking the same people across multi-year periods, we are able to ascertain whether extreme attitude holders exhibit less change in policy attitudes than moderates. The results revealed that extreme attitude holders are more likely to change their attitudes than moderates across various policy issues and general ideologies, and tend to directionally moderate over time. A final experiment finds that lay people incorrectly believe that extreme attitudes holders are more resistant to change, contrary to the results found here. We discuss the implications of this finding for understanding the evolution of extreme attitude holders, the misperception of ideological and policy differences, and the role of inaccurate out-group perceptions in shaping polarization and intergroup conflict.

---

[1] INSEAD, Fontainebleau, France. [2] University of Lübeck, Lübeck, Germany. [3] Tilburg University, Tilburg, Netherlands. ✉email: nadav.klein@insead.edu; olga.stavrova@uni-luebeck.de

In matters of politics, people with stances on the edges of the distribution—extreme attitude holders—receive substantial attention. Recent academic studies and national polls observed increasing numbers of extremists at either end of the political spectrum, a trend described as a sign of political polarization[1,2]. Scholars have identified the problematic aspects of this trend, suggesting that it aggravates intergroup conflict and ultimately harms democratic institutions[3–8]. Laypeople appear to share these concerns[9,10]. The alarm about polarization is particularly acute because it often implies worsening attitudes towards outgroups, which have reached a multi-year nadir[11]. The major worry is that people who lie at the extremes will remain stubbornly attached to their views and resistant to change[12], resulting in an increasingly intractable intergroup conflict.

But are extreme attitude holders indeed more likely to resist change and ossify in their views than moderates? Mapping out individual developmental trends in political views is important for understanding the role of extreme and moderate opinion holders in polarization trends. Yet, existing research on temporal patterns in polarization has used pooled cross-sectional data that do not track the same people over time. In the current studies, we used longitudinal data surveying the same individuals for over a decade to provide a clearer view on the evolution of extreme attitude holders. Since the ultimate purpose of political discourse is influencing policy choices, we focus on attitudes related to public policies.

The picture of extreme attitude holders as being less likely to change than moderates might seem appealing. When asked to define extremism, laypeople tend to mention unwillingness to listen to others, inflexibility, and close-mindedness[13]. It is easy to bring to mind an image of extremists who write scathing letters to the editor, go on the streets to protest, or join radical groups. Such archetypes may seem unlikely to moderate their views over time.

However, there are also valid reasons to suggest that extremists might moderate over time. First, the reasons for adopting extreme attitudes are varied and while some of them are compatible with attitude stability, others are compatible with attitude change. Some people adopt extreme positions because of relatively intransient factors such as ideological fervor or deeply held moral convictions. However, others do because of more transient factors, such as idiosyncratic life events or social influence. For example, new parents might become tough on crime because of the additional concern for safety that having children naturally brings (ref. [14], Study 1). Others adopt extreme political attitudes due to parental influence[15] which may wane over time. Still others adopt extreme attitudes as a way of signaling to others their moral values rather than actually having these moral values (virtue-signaling; ref. [16]). Overall, there are several reasons that lead people to adopt extreme attitudes, and not all of them imply increased resistance to change.

Second, people can in fact be persuaded by arguments and information from the other side. Although some research suggests that motivated reasoning processes can lead people to respond defensively to counter-attitudinal information and sometime end up with even more polarized attitudes about specific policy issues[17–19], other evidence finds that people can also logically incorporate counter-attitudinal arguments and moderate their policy positions[20–22]. Motivated reasoning processes, while powerful, are not without bounds: People will generally interpret information in ways that support their favored conclusions only when such interpretations are reasonably justifiable[23]. Thus, although they may resist it, people *can* be influenced by counter-arguments and this in turn may mean that some extremists can moderate their policy positions.

Third, existing research suggests that mechanistically explaining policy issues can make people adopt more moderate attitudes[24]. It is possible that learning more about issues over time can lead extreme attitude holders to moderate their positions. This possibility is consistent with work suggesting that information acquisition can lead to attitude change[25] and that attitude change tends to occur more frequently among younger people[26] who are in the process of acquiring knowledge and building the capability to explain the nuances of policy issues.

Fourth, it is possible the extreme attitude holders identify and follow their chosen political parties more strongly than moderates[27]. This in turn may mean that extreme attitude holders would toe the party line even when their chosen political party changes directions or makes compromises as a way of expanding its electoral reach.

Finally, people might adopt extreme attitudes to stand out from the mainstream. Indeed, political extremists tend to score lower on conformity traits[28]. Since what is considered mainstream can change over time, extremists might abandon their extreme positions on the issues on which they anticipate their (originally extreme) views to become more conventional, common, and widely-endorsed in the near future (the same way early adopters abandon fashion items that become mainstream).

Existing research on attitude extremity provides only mixed evidence about its link with attitudes stability. Some studies have found that attitude extremity predicted stability[29,30], whereas another did not[31]. Still another study found that attitude extremity predicted stability for some attitudes but not others[32]. It is possible that some of these inconsistencies can be reconciled by broadening the scope of the previous studies, which measured attitudes over short time periods. In other words, although extremists may be more resistant to change than moderates in the short run (i.e., a period of weeks, ref. [30]), this conclusion may be different once measured in the long(er) run. We therefore build on the previous studies by examining attitude stability over multi-year periods of up to 13 years and by using more than two measurement points. This long-term perspective is important because it provides a lens for understanding polarization in policy attitudes and allows us to test one aspect of the assumption of a disappearing center whereby moderates are assumed to drift towards the extreme ends over time[33].

We use the General Social Survey (GSS) from the United-States and the Longitudinal Internet Studies for the Social Sciences (LISS) from the Netherlands, enabling us to track people who were either extreme or moderate in their attitudes in earlier periods and measure the degree of attitude change they evinced in later periods. We test two questions First, whether extreme attitude holders are more or less likely to change policy attitudes than moderates; and second, when extreme attitude holders do exhibit attitude change, whether the direction of change tends towards the extreme or the middle. Following this, we provide additional analyses that test whether measuring general political ideology rather than specific policy issues yields similar results. Finally, we report an experimental study that tested whether people's intuitions about change exhibited by extreme and moderate attitude holders align with actual attitude change.

## Methods

**Study 1**. Study 1 examined the effect of attitude extremity on attitude change over time. We used the General Social Survey (GSS), a large-scale nationally representative data on attitudes about a variety of policy and socio-political issues collected from American adults over a period of 6 years. We compared attitude change exhibited over this time by people who held a more extreme (e.g., extreme support or extreme opposition) attitude towards policy issues to more moderate attitude holders. For purposes of replicability, we adopted the Exploring Small,

Confirming Big analytic strategy[34], which involves randomly splitting the data into an exploratory part used to develop hypotheses and a confirmatory part used to test these hypotheses in out-of-sample data. Pre-registration is at https://osf.io/h67bk.

We used the three panel datasets collected within the General Social Survey, GSS[35] Each panel dataset consists of unique GSS participants who have been invited to join the panel study in 2006, 2008, or 2010. Each dataset includes three waves separated by a two-year time lag (e.g., for participants who joined in 2006, wave 2 was in 2008 and wave 3 was in 2010; for participants who joined in 2008, wave 2 was in 2010 and wave 3 in 2012 etc.). The datasets include multiple questions assessing attitudes towards different policy and socio-political issues. For the present analyses, we selected all attitude items that were measured in all three waves using a continuous scale, which is necessary for assessing extreme vs. moderate stances about an issue.

Since all three panel datasets used the same design (2-year lag) and measures, we combined them into one ($n = 6067$). All attitude items included response options "Don't know" and "No answer" that we coded as missing (only 0.1% to 3.4% of respondents selected one of these options). We removed cases that had missing values on either all of the attitude items or who participated only in the first wave, resulting in the final dataset of $n = 4668$.

Following Exploring Small, Confirming Big analytic strategy[34], we randomly split the dataset into an exploratory and a confirmatory (or hold-out) part. Twenty percent of the data constituted the Exploratory Sample ($n = 933$, $M_{\text{age wave 1}} = 47.38$, $SD_{\text{age wave 1}} = 16.99$; 431 male, 502 female based on information provided by participants) and 80% constituted the Hold-Out Sample, used for confirmatory analyses ($n = 3,735$, $M_{\text{age wave 1}} = 47.75$, $SD_{\text{age wave 1}} = 17.09$; 1587 male, 2148 female based on information provided by participants). We present the results of both exploratory and confirmatory analyses separately.

The survey included measures of participants' attitudes towards the following policy and socio-political issues: gender egalitarianism, support for affirmative action, support for redistribution, support for gays/lesbians' rights, approval of corporal punishment for children, permissive sex attitudes and justification of infidelity. See Supplementary Table 1 for the scale items, response options, and internal consistencies.

We combined some scale items by averaging them into a composite where semantically appropriate. This helped mitigate concerns about random measurement error inherent to all surveys, because averaging across several questions helps reduce the overall random error that might have afflicted each separate question on its own and increases the chance that the composite provided an indication of the true attitudes of the survey participants.

We conducted two types of analyses. First, to answer our first research question – whether extremists are less likely to change policy attitudes than moderates – we regressed within-individuals attitude fluctuation over-time on the linear and quadratic terms of the respective attitude at baseline. To measure within-individuals attitude fluctuation over time, we computed the Mean Square of Successive Differences (MSSD[36]) for each attitude for each participant over the three waves (see ref. [37] for a similar approach). MSSD is a dimensionless measure of variability over time as it takes out variability due to gradual shifts in the mean over time. A larger MSSD denotes stronger fluctuations in an individual's responses between the successive time points, with the minimum possible MSSD of 0 indicating the same scale response over time (complete attitude stability). This analysis was not pre-registered because it occurred to us to do it after feedback on our pre-registered analysis.

In an alternative specification that was pre-registered, we operationalized variability using standard deviation computed for each participant over the three waves. Similar to MSSD, this measure reflects individual differences in the over-time variability in attitudes (higher values = more variability), yet in contrast to MSSD it does not take into account potential gradual shifts in the individual's mean. Both measures of variability yielded substantively similar results.

Second, to answer our second research question – whether the direction of change in extremists (vs. moderates) tends towards the extreme or the middle – we analyzed the attitude trajectory over time for each participant as a function of the extremity of the participant's initial attitude. This analysis was not pre-registered because this research question had not occurred to us at the time of pre-registration.

Specifically, we tested whether the effect of time on each attitude is non-linearly (i.e., quadratically) moderated by individuals' attitudes at baseline. Given the nested nature of the data (each individual's attitudes were measured multiple times), we conducted multilevel regressions with random intercepts for each participant. We tested the following quadratic moderation model:

$$\hat{y} = i_Y + b_1 X + b_2 X^2 + b_3 W + b_4 XW + b_5 X^2 W \qquad (1)$$

where $\hat{y}$ is the predicted attitude value at each time point, X is time and W is the respective attitude value at baseline. An interaction effect denoted by $b_5$ ($X^2 W$) would indicate that individuals with extreme (both low and high) attitudes at baseline have different developmental trajectories over time compared to individuals with moderate attitudes at baseline. To visualize this interaction, we plotted the trajectory of attitude change over time experienced by individuals with high, moderate and low attitude values at baseline.

Finally, as a robustness test, we also tested whether classifying participants by general political ideology yields similar results to those seen using specific socio-political issues. The GSS dataset measures ideology on a scale ranging from 1 (extremely liberal) to 7 (extremely conservative) with labels attached to all scale-points (i.e., 2 = liberal; 3 = slightly liberal; 4 = moderate, middle of the road, etc.). We used this item to conduct this test.

Data distribution was assumed to be normal but this was not formally tested.

**Study 2.** Study 2 used a different set of attitudes, data from a different country (Netherlands), a larger sample (~13,000 individuals), and longer time span (up to 13 years). Pre-registration is at https://osf.io/8ytev. Because this dataset contains 13 waves of surveys, it provides another opportunity to mitigate concerns about regression to the mean: Due to random measurement error inherent to all surveys, some attitudes measured as extreme in time 1 may indeed belong to extreme attitude holders or, alternatively, may reflect a particularly large measurement error. If the latter, then what appears to be moderation in time 2 might simply be a regression to the true attitude of the opinion holder due to a smaller measurement error at subsequent measurement points. The dataset in Study 2 helps here by having numerous survey waves (13 waves vs. only 3 waves in Study 1), which increases the number of measurements and thus reduces measurement error across all of them[38]. As in Study 1, we group some of the variables into composites where semantically appropriate, which also helps reduce measurement error in any one of these variables (scale reliability statistics are presented in Supplementary Table 5).

As for our main hypotheses, we again test whether extreme supporters or opposers of an issue are less likely to change their attitude over time, compared to individuals who held a more moderate attitude initially. We again randomly split the data into an exploratory part and a confirmatory part. Following the results

of Study 1 and our exploratory analysis of 20% of the data, we did not expect extreme attitudes to be more stable over time.

We used the data from the Longitudinal Internet Studies for the Social Sciences, LISS Panel[39]. The panel consists of a large nationally representative sample of the Dutch population. The panel started in 2008 and at the time of writing has accomplished 13 waves (last wave in 2021). Panel participants are asked to respond to short surveys (referred to as 'modules') on different topics monthly, such that each survey (or module) is repeated annually. We used the module "Politics and Values" that included multiple questions assessing participants' attitudes towards different policy and socio-political issues. Our analyses included all policy and socio-political attitude items that were repeatedly assessed in the module.

The entire dataset included 15,561 participants. We retained the cases with valid values on attitude items in at least 2 waves ($n = 11,570$). As in Study 1, we randomly split the data into the Exploratory sample (20%) and Holdout sample (80%). The Exploratory dataset consisted of 2260 participants ($M_{\text{age wave 1}} = 42.26$, $SD_{\text{age wave 1}} = 18.86$; 1053 male, 1207 female based on information provided by participants). The Holdout sample consisted of 9310 participants ($M_{\text{age wave 1}} = 42.31$, $SD_{\text{age wave 1}} = 17.92$; 4208 male, 5102 female based on information provided by participants).

We included the attitudes towards a variety of policy and socio-political issues, ranging from support of euthanasia to gender egalitarianism to multiculturalism. The attitudes items, measures, response options and internal consistencies are provided in Supplementary Table 5. Some of the items included the response option "Don't know" that was coded as missing. We used the same analytic strategy as in Study 1.

Finally, as a robustness test, we tested whether classifying participants by general political ideology yields similar results to those seen using specific socio-political issues The LISS dataset measures ideology on a scale ranging from 0 (left) to 10 (right), and we used this item to conduct this test.

Data distribution was assumed to be normal but this was not formally tested.

**Study 3**. Study 3 tested whether people hold a lay belief in stubborn extremists (i.e., a belief that extremists are less susceptible to attitude change than moderates) in an experiment. Participants were presented with the attitudes of different target persons on various policy issues. We manipulated whether those target persons were on the extreme end of stances or were moderate, and measured participants' predictions about how likely these target persons were to change their attitudes in the future. We also tested whether the results held regardless of participants' own opinions on the various policy issues presented.

We obtained IRB approval (ref: 2021-63) for this experiment and participants provided informed consent. Without past indication of expected effect size, we used the rule of thumb of recruiting at least 75 participants per between-subjects experimental cell. A sensitivity power analysis showed that this sample size would allow us to detect the difference between the conditions (extreme vs. moderate target) of at least $d = 0.32$ with 80% power and alpha of 0.05. We have not analyzed any data until collection was completed. This study was pre-registered at https://aspredicted.org/KPG_Q49.

We recruited 300 participants from Cloud Research and 17 failed the attention check, resulting in $N = 283$ ($M_{\text{age}} = 46.00$, $SD_{\text{age}} = 13.73$; 136 women, 146 men, 1 other based on information provided by participants). Compensation was $0.48. This was a 2(Attitude: moderate vs. extreme) x 2(Stance: oppose vs. support) x 3(Policy issue: affirmative action, wealth redistribution, gay marriage) mixed design, with the first two factors manipulated between and the third factor manipulated within subjects. We were interested in the effect of the first factor (Attitude: moderate vs. extreme) and varied the other two factors (stance and policy issues) to increase the generalizability of the findings.

Participants read that there had been a wide-ranging survey of opinions about various policy issues and that they would see several of those people's answers to questions about these policy issues. Participants were then presented with three target persons, each of whom purportedly responded to a policy question. Information about each of the three target persons was presented on a different screen, with order counter-balanced.

We sampled three policy issues taken from the General Social Survey (GSS) used in Study 1: affirmative action, government-sponsored wealth redistribution, and gay marriage. For each policy issue, participants read the question taken verbatim from the GSS and read the response of a target person. This response was manipulated to be either in support or in opposition to the policy issue, and critically, to be either extreme or moderate in its support or opposition as detailed below.

For affirmative action, the original GSS survey question allowed for a response on a 4-point scale ranging from "strongly in favor" to "strongly opposed." Participants read that a target person ("Mary") responded that she either extremely favors or opposes affirmative action (options 1 or 4 on the scale, respectively) or that she either moderately favors or opposes affirmative action (options 2 or 3 on the scale, respectively).

For wealth redistribution, the original GSS survey question allowed for a response on a 7-point scale ranging from "the government should [redistribute wealth]" to "the government should not [redistribute wealth]." Participants read that a target person ("John") responded that he either extremely favors or opposes wealth redistribution (options 1 or 7 on the scale, respectively) or that he either moderately favors or opposes wealth redistribution (options 3 or 5 on the scale, respectively).

For gay marriage, the original GSS survey question allowed for a response on a 5-point scale ranging from "strongly agree" to "strongly disagree." Participants read that a target person ("Judy") responded that she either extremely favors or opposes gay marriage (options 1 or 5 on the scale, respectively) or that she either moderately favors or opposes gay marriage (options 2 or 4 on the scale, respectively).

For each target person, participants were asked whether they believed that s/he is likely to change his/her opinion about the policy issue (1 = extremely unlikely; 7 = extremely likely). After completing responses for all three target persons, participants were asked for their own stances on these three policy issues using the original GSS questions. Participants were also asked for their own general political attitudes (1 = very conservative/right-wing; 7 = very liberal/left-wing). Data distribution was assumed to be normal but this was not formally tested.

**Reporting summary**. Further information on research design is available in the Nature Portfolio Reporting Summary linked to this article.

## Results
**Study 1**. To test whether extremists exhibit less attitude change than moderates, we assessed whether the association between attitude score at wave 1 and within-individuals over-time fluctuations in that attitude (MSSD) follows an inversed u-shaped relationship. Such a relationship would imply that individuals who held extreme attitudes (support or opposition) at wave 1

exhibited less change (i.e., lower MSSDs) over time relative to individuals who held mode moderate attitudes at wave 1.

For each attitude item, we regressed attitude fluctuation (MSSD) on the respective attitude score at wave 1 (mean-centered) and its quadratic term. The model coefficients obtained in the Exploratory and the Holdout samples are shown in Table 1. The quadratic term reached significance for 7 (out of 8) attitudes in the exploratory sample and for all 8 attitudes in the holdout sample. It was positive for gender egalitarianism, support for affirmative action, support for redistribution, support for contraception for teens, disapproval of child corporal punishment, permissive sex attitudes, and justification of sexual infidelity, $bs \geq 0.08$, $ps \leq 0.001$, and negative for support of gays/lesbians, $b = -0.16$, $p < 0.001$, 95% CI[−0.21, −0.11]. We replicate this analysis when measuring attitude fluctuation using standard deviation in Supplementary Table 2.

The shape of this non-linear effect is shown in Fig. 1 for the Holdout sample (see Supplementary Fig. 1 for the Exploratory sample). Out of 8 attitudes, only one (support for gay/lesbian rights, which had a negative quadratic effect) followed the inversed-u-shaped pattern, with extreme attitude holders at both ends exhibiting less fluctuations than moderates. For five attitudes, we found a u-shaped pattern, with extremists exhibiting *more* (rather than less) attitude change over time and for two remaining attitudes, we found no distinct u-shape pattern.

Next, we explored the direction of change of extremists and moderates. We modeled the attitude trajectory over time for each participant as a quadratic function of the participant's initial attitude. We tested whether the effect of time on each attitude is non-linearly (i.e., quadratically) moderated by individuals' attitudes at baseline. The results are presented in Table 2. The interactions between the quadratic effect of time and baseline attitude, $bs \geq 0.10$, $ps \leq 0.001$, suggest that extremists differed in their attitude development from moderates. To visualize the pattern of this interaction, we plotted the trajectory of attitude change over the three years experienced by individuals who held extremely favorable, extremely opposing, or moderate attitudes towards each issue at wave 1. For policy issues measured by multi-item scales, extreme was classified as top or bottom 10% of the distribution of attitude holders and moderate was classified as the rest (the 20–80% middle). The results (Fig. 2 for the Holdout sample and Supplementary Fig. 2 for the Exploratory sample) illustrate that extremists at both ends of the distribution converged in their attitudes towards the mean over time, while moderates barely changed.

To further address the possibility that a measurement error affected our results, we conducted an additional non-preregistered analysis. Specifically, we tested how attitude items changed over time relative to one another. If the effects we find reflect true attitude change, then attitude items that are closely related to each other (e.g. permissive sex attitudes and support of gays/lesbians) would also be more likely to change together and exhibit the same pattern of change in later waves of the data. In contrast, unrelated attitude items should not change together over time and instead exhibit different patterns of change in later waves of the data. Put more simply, closely related attitudes should change together and unrelated attitudes should not change together. In contrast, if the effects we find reflect regression to the mean or measurement error, then even attitude items that are unrelated to each other should change together and exhibit a similar pattern of change over time. This is because a measurement error would not discriminate between related and unrelated attitudes – it would affect everything.

We first computed a similarity score by correlating each pair of attitude items in our data. Higher correlations reflect greater attitude similarity. We next computed a change score by subtracting attitude at the last wave from the attitude value at the first wave ([attitude$_{first}$ – attitude$_{last}$]). Higher values indicate greater decrease over time. Supplementary Fig. 3 plots the relationship between attitude similarity and attitude change, showing that it is strongly positive ($n = 72$ issue pairs, $r^{70} = 0.82$, $p < 0.0001$, 95% CI[0.73, 0.88]). This means that similar attitudes exhibited the same change pattern whereas dissimilar attitudes exhibited different change patterns, arguing against regression to the mean or a measurement error.

To account for the possibility that the bounded scales in datasets influenced the effects, we also re-did our regressions using Tobit models that allow obtaining unbiased coefficients in the presence of floor or ceiling effects[40]. The results (Supplementary Tables 3 and 4) support the conclusions drawn from other analyses. Beyond the Tobit analyses, notice that the general results of this study (and Study 2 as well) also weaken the possibility that the results are an artifact of the bounded measurement scale. If this were the case, we would have found that extremists exhibit greater attitude stability than moderates whereas we in fact find the opposite.

**Table 1 Study 1, Exploratory and Holdout samples: Results of quadratic regression models with attitude at wave 1 (linear and quadratic terms) predicting attitude change over time (measured using *MSSD*).**

| Attitude | | **Exploratory sample** | | | **Holdout sample** | | |
|---|---|---|---|---|---|---|---|
| | | ***b*** | **95% CI** | ***p*** | ***b*** | **95% CI** | ***p*** |
| Gender egalitarianism | Linear | −0.00 | −0.06-0.06 | 0.999 | −0.01 | −0.04 to 0.01 | 0.304 |
| | Quadratic | 0.18 | 0.12-0.25 | <0.001 | 0.14 | 0.12-0.17 | <0.001 |
| Support for affirmative action | Linear | 0.22 | 0.11-0.33 | <0.001 | 0.17 | 0.11-0.23 | <0.001 |
| | Quadratic | 0.19 | 0.10-0.28 | <0.001 | 0.08 | 0.03-0.12 | 0.001 |
| Support for redistribution | Linear | 0.21 | −0.01-0.44 | 0.066 | 0.18 | 0.06-0.29 | 0.002 |
| | Quadratic | 0.20 | 0.08-0.32 | 0.001 | 0.21 | 0.15-0.27 | <0.001 |
| Support for contraception for teens | Linear | 0.10 | −0.03-0.23 | 0.121 | 0.08 | 0.02-0.14 | 0.007 |
| | Quadratic | 0.30 | 0.17-0.44 | <0.001 | 0.24 | 0.18-0.30 | <0.001 |
| Support of gays/lesbians | Linear | −0.03 | −0.08-0.03 | 0.321 | −0.00 | −0.03 to 0.03 | 0.936 |
| | Quadratic | −0.14 | −0.22-−0.06 | 0.001 | −0.16 | −0.21 to −0.11 | <0.001 |
| Disapproval of child corporal punishment | Linear | −0.01 | −0.08-0.07 | 0.870 | −0.01 | −0.05 to 0.04 | 0.689 |
| | Quadratic | 0.19 | 0.12-0.25 | <0.001 | 0.16 | 0.12-0.20 | <0.001 |
| Permissive sex attitudes | Linear | 0.02 | −0.06-0.10 | 0.655 | 0.04 | 0.00-0.08 | 0.043 |
| | Quadratic | 0.16 | 0.08-0.25 | <0.001 | 0.11 | 0.06-0.15 | <0.001 |
| Justification of sexual infidelity | Linear | 0.46 | 0.16-0.76 | 0.002 | 0.13 | −0.02-0.27 | 0.096 |
| | Quadratic | 0.09 | −0.10-0.29 | 0.357 | 0.33 | 0.25-0.41 | <0.001 |

*N* = 4668.

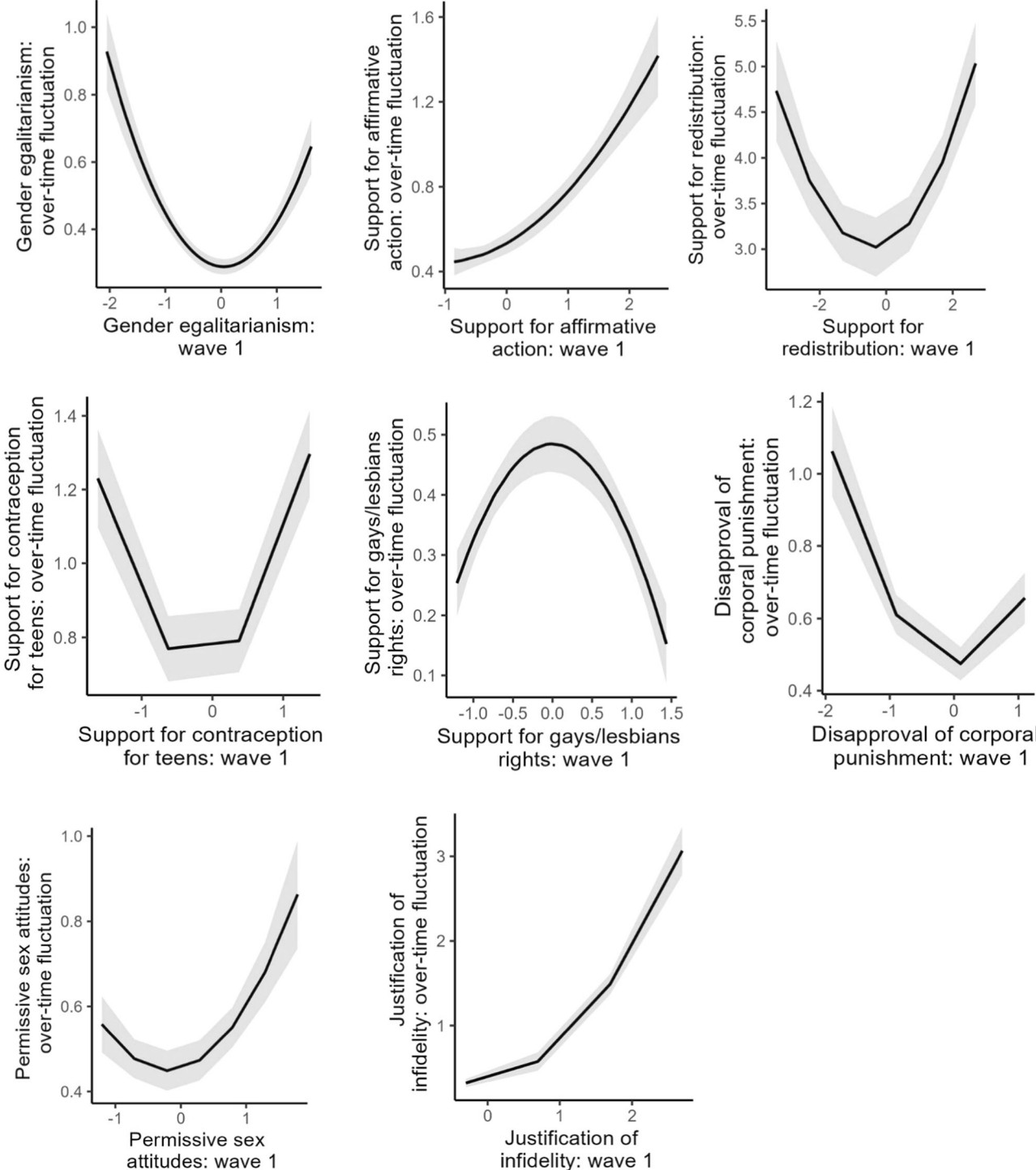

**Fig. 1 Association between attitude at wave 1 and attitude fluctuation over 3 waves (measured as within-person MSSD; the higher the value, the more fluctuation there was), Holdout sample, Study 1 (General Social Survey).** Attitudes at wave 1 were centered around mean; over-time fluctuation is represented by MSSD (higher values reflect more fluctuations, 0 reflects complete stability). $N = 3735$.

Finally, establishing robustness, we analyzed general political ideology. Table 3 and Fig. 3 show that the results were similar to those obtained with specific socio-political issues, whereby extreme ideologs exhibited more fluctuation in their attitudes compared to moderates. Specifically, the quadratic term associated with general ideology was positive in a regression model, $b = 0.23$, 95% CI[0.19, 0.26], $p < 0.001$, and the interaction

between the quadratic effect of time and baseline ideology was positive in a quadratic moderation analysis, $b = 0.18$, 95% CI[0.15, 0.20], $p < 0.001$.

In summary, Study 1 finds that compared to moderates and for most attitudes, extreme attitude holders are more (not less) likely to exhibit change over time. In general, the direction of extreme attitude holders' change is towards the middle.

**Table 2 Study 1, Exploratory and Holdout samples: Quadratic moderation analyses, Study 1.**

| Attitude | | Exploratory sample | | | Holdout sample | | |
|---|---|---|---|---|---|---|---|
| | | b | 95% CI | p | b | 95% CI | p |
| Gender egalitarianism | Time linear | 0.00 | −0.02 to 0.02 | 0.981 | −0.01 | −0.02 to −0.00 | 0.026 |
| | Time quadratic | −0.01 | −0.05 to 0.03 | 0.544 | 0.01 | −0.01 to 0.02 | 0.574 |
| | Baseline attitude | 0.57 | 0.52-0.62 | < 0.001 | 0.62 | 0.60-0.65 | < 0.001 |
| | Time linear x baseline attitude | −0.23 | −0.26 to −0.19 | < 0.001 | −0.20 | −0.22 to −0.19 | < 0.001 |
| | Time quadratic x baseline attitude | 0.20 | 0.14-0.26 | < 0.001 | 0.17 | 0.14-0.20 | < 0.001 |
| Support for affirmative action | Time linear | −0.03 | −0.06 to 0.00 | 0.066 | 0.01 | −0.00 to 0.03 | 0.153 |
| | Time quadratic | 0.00 | −0.05 to 0.05 | 0.994 | −0.02 | −0.04 to 0.01 | 0.207 |
| | Baseline attitude | 0.53 | 0.48 to 0.58 | < 0.001 | 0.58 | 0.55-0.60 | < 0.001 |
| | Time linear x baseline attitude | −0.20 | −0.24 to −0.17 | < 0.001 | −0.23 | −0.24 to −0.21 | < 0.001 |
| | Time quadratic x baseline attitude | 0.26 | 0.21-0.32 | < 0.001 | 0.20 | 0.17-0.23 | < 0.001 |
| Support for redistribution | Time linear | −0.08 | −0.15 to −0.01 | 0.021 | −0.08 | −0.12 to −0.05 | < 0.001 |
| | Time quadratic | 0.10 | −0.02 to 0.22 | 0.091 | −0.02 | −0.09 to 0.04 | 0.437 |
| | Baseline attitude | 0.48 | 0.43-0.53 | < 0.001 | 0.53 | 0.51-0.56 | < 0.001 |
| | Time linear x baseline attitude | −0.23 | −0.26 to −0.19 | < 0.001 | −0.22 | −0.24 to −0.21 | < 0.001 |
| | Time quadratic x baseline attitude | 0.30 | 0.24-0.36 | < 0.001 | 0.25 | 0.22-0.28 | < 0.001 |
| Support for contraception for teens | Time linear | −0.02 | −0.06 to 0.02 | 0.305 | −0.02 | −0.04 to −0.00 | 0.016 |
| | Time quadratic | −0.01 | −0.08 to 0.06 | 0.786 | 0.01 | −0.02 to 0.04 | 0.488 |
| | Baseline attitude | 0.55 | 0.49-0.60 | < 0.001 | 0.60 | 0.57-0.63 | < 0.001 |
| | Time linear x baseline attitude | −0.27 | −0.30 to −0.23 | < 0.001 | −0.25 | −0.27 to −0.24 | < 0.001 |
| | Time quadratic x baseline attitude | 0.20 | 0.13-0.26 | < 0.001 | 0.16 | 0.13-0.19 | < 0.001 |
| Support of gays/lesbians | Time linear | −0.02 | −0.04 to 0.00 | 0.070 | −0.01 | −0.02 to −0.00 | 0.037 |
| | Time quadratic | −0.01 | −0.05 to 0.03 | 0.753 | 0.00 | −0.02 to 0.02 | 0.842 |
| | Baseline attitude | 0.81 | 0.77-0.85 | < 0.001 | 0.79 | 0.77-0.81 | < 0.001 |
| | Time linear x baseline attitude | −0.11 | −0.14 to −0.09 | < 0.001 | −0.12 | −0.13 to −0.10 | < 0.001 |
| | Time quadratic x baseline attitude | 0.08 | 0.03-0.12 | < 0.001 | 0.10 | 0.08-0.12 | < 0.001 |
| Disapproval of children corporal punishment | Time linear | −0.03 | −0.06 to −0.00 | 0.036 | −0.02 | −0.04 to −0.01 | 0.004 |
| | Time quadratic | −0.04 | −0.08-0.01 | 0.147 | −0.01 | −0.03 to 0.02 | 0.458 |
| | Baseline attitude | 0.59 | 0.54-0.64 | < 0.001 | 0.58 | 0.55-0.61 | < 0.001 |
| | Time linear x baseline attitude | −0.24 | −0.28 to −0.21 | < 0.001 | −0.21 | −0.23 to −0.20 | < 0.001 |
| | Time quadratic x baseline attitude | 0.17 | 0.11-0.22 | < 0.001 | 0.21 | 0.18-0.24 | < 0.001 |
| Permissive sex attitudes | Time linear | 0.00 | −0.03 to 0.03 | 0.872 | 0.02 | 0.01-0.04 | 0.001 |
| | Time quadratic | −0.00 | −0.05 to 0.04 | 0.867 | 0.00 | −0.02 to 0.03 | 0.832 |
| | Baseline attitude | 0.63 | 0.59-0.68 | < 0.001 | 0.66 | 0.63-0.68 | < 0.001 |
| | Time linear x baseline attitude | −0.17 | −0.20 to −0.14 | < 0.001 | −0.18 | −0.20 to −0.16 | < 0.001 |
| | Time quadratic x baseline attitude | 0.20 | 0.14-0.25 | < 0.001 | 0.16 | 0.14-0.19 | < 0.001 |
| Justification of sexual infidelity | Time linear | 0.02 | −0.01 to 0.04 | 0.235 | −0.00 | −0.01 to 0.01 | 0.946 |
| | Time quadratic | −0.01 | −0.06 to 0.03 | 0.502 | 0.01 | −0.01 to 0.03 | 0.469 |
| | Baseline attitude | 0.50 | 0.44-0.56 | < 0.001 | 0.41 | 0.38-0.44 | < 0.001 |
| | Time linear x baseline attitude | −0.26 | −0.30 to −0.22 | < 0.001 | −0.30 | −0.32 to −0.28 | < 0.001 |
| | Time quadratic x baseline attitude | 0.25 | 0.18-0.32 | < 0.001 | 0.29 | 0.26-0.33 | < 0.001 |

All predictors were centered; all models included a random intercept at the level of participants. $N = 4668$.

**Study 2.** To test whether extremists exhibit less attitude change than moderates, for each attitude we regressed attitude over-time fluctuation (MSSD) on the respective attitude score at wave 1 (mean-centered) and its quadratic term. The model coefficients from the exploratory and the holdout sample are shown in Table 4. The quadratic term was significant and positive for all attitudes in both the Exploratory and the Holdout sample, $bs \geq 0.04$, $ps \leq 0.001$, with the exception of justification of euthanasia, $b = −0.07$, $p < 0.001$, 95% CI[−0.08, −0.05]. We replicate this analysis when measuring attitude fluctuation using standard deviation in Supplementary Table 6.

The shape of the association between attitude at wave 1 and attitude change is shown in Fig. 4 for the Holdout sample (see Supplementary Fig. 4 for the Exploratory sample). For all attitudes but one (euthanasia), we detected a pattern where extremists exhibited more attitude fluctuations over time than moderates. Note that for some attitudes, extremists at both ends of the spectrum showed a relatively similar magnitude of fluctuations, while for others (especially, support for immigration, gender egalitarianism at work and support for income equality), conservative extremists were most volatile, followed by liberal extremists and moderates who showed the most attitude stability.

To test whether extremists change towards the extreme or the middle, we tested whether the effect of time on each attitude is non-linearly (i.e., quadratically) moderated by individuals' attitudes at baseline. The results are presented in Table 5. For

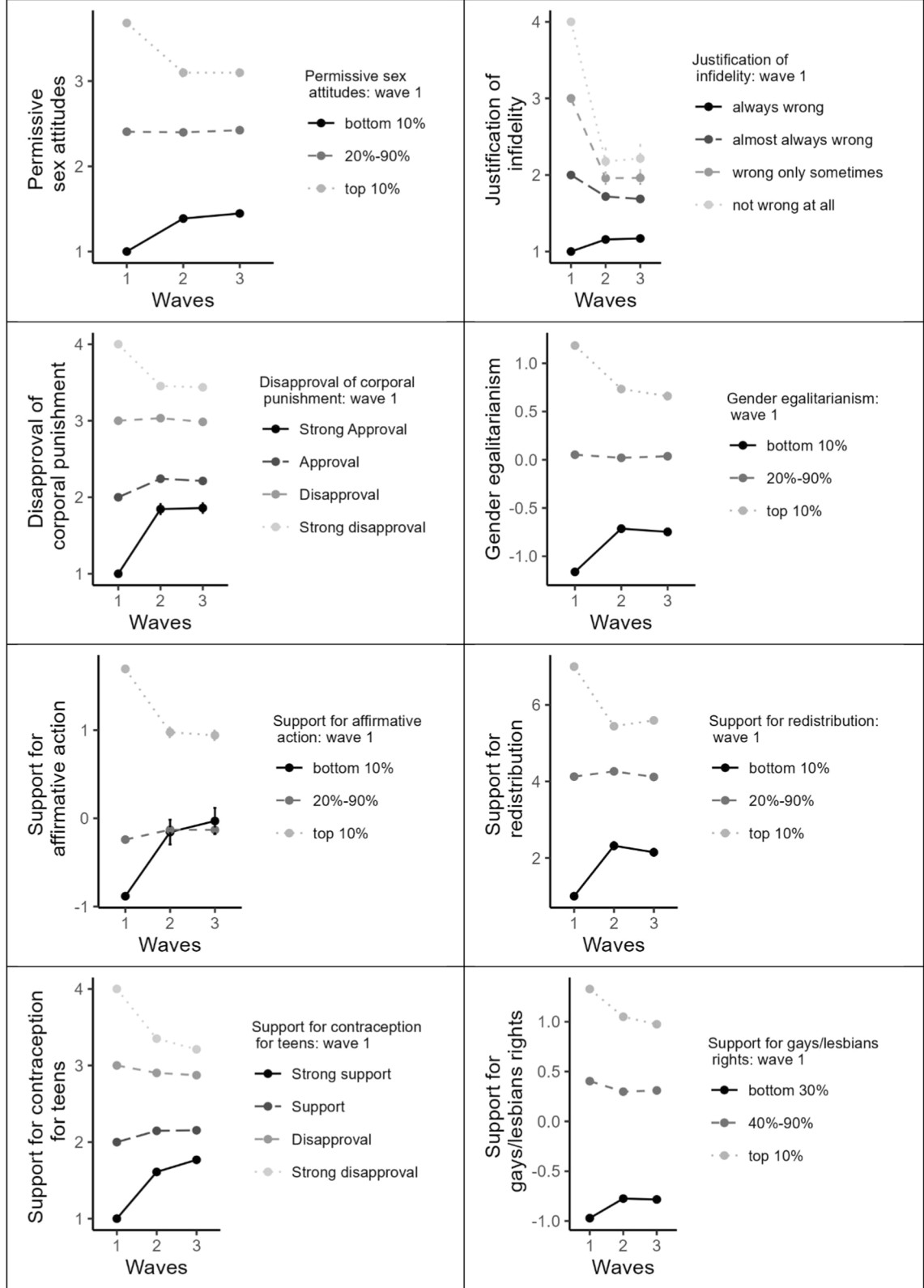

**Fig. 2 Attitude change trajectories as a function of initial attitude (General Social Survey), Study 1, Holdout sample.** For single-item constructs that used 4-point response scale we present the over-time trajectories for each response category in wave 1, rather than top and bottom 10%. Support for gays/lesbians: there were only 10 participants in the bottom 10% of the distribution (due to skewness), therefore we plotted bottom 30%. Error bars are standard errors (barely visible on most graphs). $N = 3735$.

**Table 3 Results of quadratic regression models with political ideology at wave 1 (linear and quadratic terms) predicting change in political ideology over time (measured with MSSD), and quadratic moderation analyses, exploratory and holdout samples together**

**Regression model, political ideology**

| | | Study 1 | | | Study 2 | | |
|---|---|---|---|---|---|---|---|
| | | *b* | 95% CI | *p* | *b* | 95% CI | *p* |
| Political ideology | Linear | −0.01 | −0.07 to 0.05 | 0.73 | −0.04 | −0.08 to 0.003 | 0.072 |
| | Quadratic | 0.23 | 0.19–0.26 | < 0.001 | 0.13 | 0.11–0.14 | < 0.001 |
| Quadratic moderation analyses, political ideology | | | | | | | |
| Time linear | | 0.008 | −0.01 – 0.03 | 0.37 | −0.0002 | −0.003 to 0.002 | 0.84 |
| Time quadratic | | 0.01 | −0.02 to 0.04 | 0.39 | −0.0009 | −0.002 to 0.0002 | < 0.01 |
| Baseline attitude | | 0.62 | 0.60–0.64 | < 0.001 | 0.75 | 0.74–0.76 | < 0.001 |
| Time linear x baseline attitude | | −0.20 | −0.22 to −0.19 | < 0.001 | −0.02 | −0.02 to −0.02 | < 0.001 |
| Time quadratic x baseline attitude | | 0.18 | 0.15–0.20 | < 0.001 | 0.002 | 0.002–0.002 | < 0.001 |

Study 1, $N = 4668$. Study 2, $N = 11,570$. For quadratic moderation analyses, all predictors were centered and all models included a random intercept at the level of participants.

all attitudes in both the Exploratory and the Holdout sample, extremists differed in their attitude development from moderates as evidenced by the interactions between the quadratic effect of time and baseline attitudes, $bs \geq 1.564\text{e-}03$, $ps \leq 0.001$. Notice that although these coefficients are small, the standard errors are commensurably small, and the best way to understand quadratic effects is to plot them. We thus visualize these results in Fig. 5 by plotting the trajectory of attitude change over the years experienced by individuals who held extremely favorable, extremely opposing, or moderate attitudes towards each issue initially (see Supplementary Fig. 5 for the Exploratory sample). For policy issues measured through multi-item scales, extreme was classified as top and bottom 10% of the distribution of attitude holders and moderate was classified as the rest (the 20–90% middle). As in Study 1, the visualizations indicated that, for most issues, extremists at both ends of the distribution tended to converge in their attitudes towards the middle over time, while moderates experienced little change.

As in Study 1, we conducted an additional non-preregistered analysis to address regression to the mean. As before, we measured the relationship between attitude similarity and attitude change. If our effects reflect true attitude change, then similar attitude items should also change together and dissimilar attitudes should change differently over time. In contrast, if our effects reflect measurement error, then even dissimilar attitudes should change together over time. Supplementary Fig. 6 plots the relationship between attitude similarity and attitude change, showing that it is strongly positive ($n = 110$ issue pairs, $r(108) = 0.91$, $p < 0.0001$. 95% CI[0.87, 0.94]). This means that similar attitudes exhibited the same change pattern whereas dissimilar attitudes exhibited different change patterns, arguing against regression to the mean or a measurement error.

In a supplementary and non-pre-registered analysis we re-did our regressions using Tobit models to account for the possibility that bounded scales influenced the effects. The results are presented Supplementary Tables 7 and 8, support our other analyses, and argue against bounded scales as a confound.

Finally, establishing robustness, we analyzed general political ideology. Table 3 and Fig. 3 show that – as in Study 1 – the results were similar to those obtained with specific socio-political issues, whereby extreme ideologs exhibited more change in their attitudes compared to moderates. Specifically, the quadratic term associated with general ideology was positive in a regression model, $b = 0.13$, 95% CI[0.11, 0.14], $p < 0.001$, and the interaction between the quadratic effect of time and baseline ideology was positive in a quadratic moderation analysis, $b = 0.002$, 95% CI[0.002, 0.002], $p < 0.001$.

In summary, Study 2 replicated the findings of Study 1 in a different country, with respect to different attitudes, and over a considerably longer period of time (13 years). As before, the results suggest that extremists tend to experience more (not less) variation in their attitudes over time compared to moderates and tend to converge towards the middle.

**Study 3.** We conducted a mixed ANOVA testing the effects of attitude (moderate vs. extreme), stance (support vs. oppose), and policy issue (affirmative action, wealth redistribution, gay marriage) with repeated measures on the third factor on perceptions of attitude change. Results revealed the critical main effect of attitude, $F(1, 279) = 137.33$, $p < 0.0001$, $\eta_p^2 = 0.330$. Across all policy issues and regardless of whether the target person opposed or supported the policy issue, participants perceived extreme attitude holders as less likely to change than moderates. Detailed results are presented in Supplementary Table 9.

The ANOVA also revealed an ancillary main effect for policy issue, $F(1, 279) = 133.03$, $p < 0.0001$, $\eta_p^2 = 0.323$ and a two-way interaction between stance and policy issue, $F(1, 279) = 50.36$, $p < 0.0001$, $\eta_p^2 = 0.153$. All other effects and interactions were nonsignificant, $Fs < 2.18$, $ps > 0.142$, $\eta_p^2 s < 0.009$.

We also tested whether the results hold when controlling for participants' own opinions on these policy issues. The critical main effect of attitude (extreme vs. moderate) remained, $F(1, 279) = 132.93$, $p < 0.0001$, $\eta_p^2 = 0.325$, suggesting that participants' perception that extremists are more likely to exhibit attitude stability than moderates did not depend on their own opinion on these political issues.

Overall, Study 3 suggests that people hold the intuition that extremists in policy attitudes are less likely to change than moderates, which stands in contrast to our findings from Studies 1-2 showing that – in reality – extremists were more likely to change than moderates.

## Discussion

Polarization is thought to weaken democratic systems and lead to intolerance[41–43]. It is therefore important to understand how extreme attitude holders and moderates evolve over time. Do extreme attitude holders tend to be set in their ways or can they moderate over time? Here we measure the time course of extremism as expressed through policy attitudes. In this context, we find that extremists have more volatile attitudes compared to moderates and that change tends to occur towards the middle.

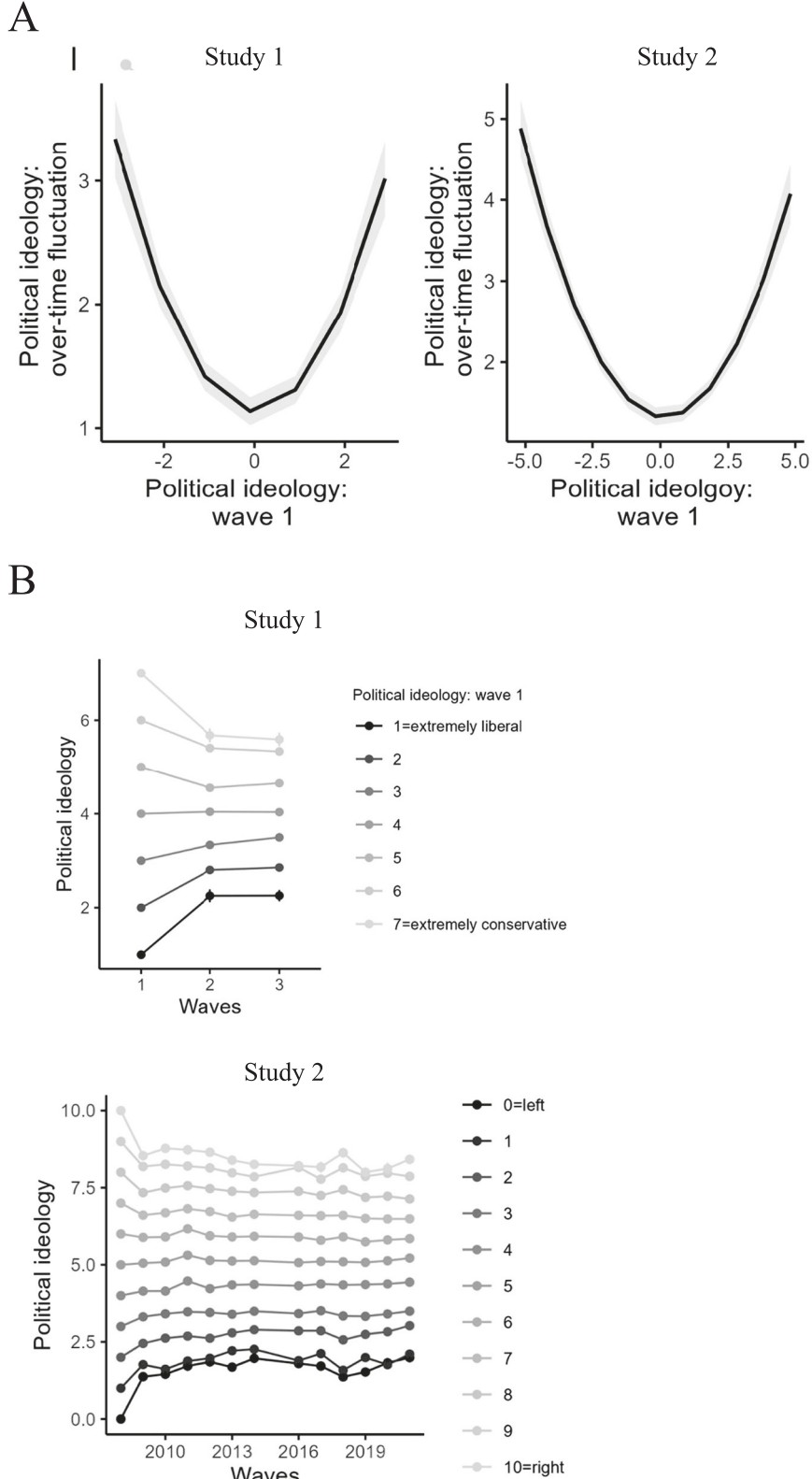

**Fig. 3 Association between political ideology at wave 1 and change in political ideology over time (Panel A) and change in political ideology as a function of initial ideology (Panel B). A** Ideology at wave 1 were centered around mean; over-time fluctuation is represented MSSD (higher values reflect more fluctuations, 0 reflects complete stability). Study 1, $N = 4668$. Study 2, $N = 11,570$. **B** Each point includes error bars that represent standard errors (too small to be discerned). Study 1, $N = 4668$. Study 2, $N = 11,570$.

**Table 4 Study 2: Results of quadratic regression models with attitude at wave 1 (linear and quadratic terms) predicting attitude change over time (measuring with MSSD).**

| Attitude | | Exploratory sample | | | Holdout sample | | |
|---|---|---|---|---|---|---|---|
| | | b | 95% CI | p | b | 95% CI | p |
| Justification of euthanasia | Linear | −0.27 | −0.33 to −0.21 | < 0.001 | −0.32 | −0.35 to −0.29 | < 0.001 |
| | Quadratic | −0.04 | −0.06 to −0.01 | 0.015 | −0.07 | −0.08 to −0.05 | < 0.001 |
| Support for income equality | Linear | −0.04 | −0.08 to 0.01 | 0.099 | −0.03 | −0.06 to −0.01 | 0.006 |
| | Quadratic | 0.09 | 0.06–0.13 | < 0.001 | 0.15 | 0.14–0.17 | < 0.001 |
| Support for acculturation | Linear | −0.06 | −0.11 to −0.01 | 0.012 | −0.06 | −0.09 to −0.04 | < 0.001 |
| | Quadratic | 0.15 | 0.12–0.18 | < 0.001 | 0.13 | 0.11–0.14 | < 0.001 |
| Disapproval of EU unification | Linear | −0.03 | −0.08 to 0.02 | 0.256 | −0.03 | −0.05 to −0.01 | 0.012 |
| | Quadratic | 0.12 | 0.09–0.16 | < 0.001 | 0.11 | 0.09–0.13 | < 0.001 |
| Gender egalitarianism in family | Linear | −0.03 | −0.05 to −0.01 | < 0.001 | −0.01 | −0.02 to −0.00 | 0.033 |
| | Quadratic | 0.06 | 0.04–0.08 | < 0.001 | 0.04 | 0.03–0.05 | < 0.001 |
| Gender egalitarianism at work | Linear | −0.02 | −0.05 to 0.01 | 0.201 | −0.04 | −0.05 to −0.02 | < 0.001 |
| | Quadratic | 0.07 | 0.03–0.10 | < 0.001 | 0.11 | 0.09–0.13 | < 0.001 |
| Multiculturalism | Linear | −0.06 | −0.07 to −0.05 | < 0.001 | −0.04 | −0.05 to −0.03 | < 0.001 |
| | Quadratic | 0.09 | 0.07–0.10 | < 0.001 | 0.06 | 0.05–0.07 | < 0.001 |
| Approval of intergenerational support | Linear | −0.04 | −0.06 to −0.01 | 0.006 | −0.01 | −0.03 to 0.00 | 0.129 |
| | Quadratic | 0.11 | 0.08–0.13 | < 0.001 | 0.13 | 0.12–0.14 | < 0.001 |
| Approval of marriage | Linear | −0.05 | −0.07 to −0.03 | < 0.001 | −0.02 | −0.03 to −0.01 | < 0.001 |
| | Quadratic | 0.05 | 0.03–0.07 | < 0.001 | 0.04 | 0.03–0.05 | < 0.001 |
| Justification of divorce | Linear | −0.06 | −0.09 to −0.02 | 0.001 | −0.02 | −0.04 to 0.00 | 0.063 |
| | Quadratic | 0.13 | 0.10–0.16 | < 0.001 | 0.12 | 0.11–0.14 | < 0.001 |

$N = 11,570$.

**Contributions**. These findings represent several contributions. First, they might help us better understand polarization. Existing research often views polarization from the lens of deteriorating attitudes towards outgroups (i.e., affective polarization)[10,11,44]. To be clear, we do not measure attitudes towards outgroups, but rather attitudes about socio-political issues. It is possible that attitudes towards outgroup are polarizing while attitudes towards specific issues are moderating.

Moreover, the evidence on affective polarization describes the evolution of groups, whereas the present analyses track the evolution of individuals. Research on attitudes towards outgroups uses pooled cross-sectional data (i.e., measuring different people over multi-year periods), finding that the people of the current generation dislike outgroups more than people of previous generations. To make inferences about specific individuals' attitudes towards outgroup, a longitudinal analysis as we do here with respect to policy attitudes will be necessary. This will help shed light on whether antipathy towards outgroups is increasing or moderating over the average person's lifespan. It is entirely possible for outgroup attitudes to be polarizing at the group level but moderating at the individual level.

The comparison with the findings on affective polarization also raises intriguing conceptual questions. Specifically, it raises the interesting possibility that changes in policy attitudes over time do not necessarily depend on the degree to which people feel antipathy towards adversaries. In other words, a person can become more moderate in matters of policy while also vehemently dislike members of the opposing party. Or, even more interestingly, a person might become more extreme in policy issues while maintaining respect towards those who disagree.

Second, a potential divergence in the trajectories of extremism in policy issues and extremism in attitudes towards outgroups may suggest that these two facets of polarization can be driven by different psychological processes. Attitudes towards policies are explained through moral values[45,46]. In contrast, attitudes towards outgroups (and ingroups) are explained through social identity theory—the sense of belonging to a social group and the motivation to bolster one's own group and derogate a competing group[47–49].

More broadly, the picture of polarization is enriched by contrasting its expression in policy attitudes versus outgroup attitudes.

Third, the present research raises future directions related to understanding why extremists tend to moderate over time, at least in policy attitudes. As we note, one type of explanation might relate to persuadability: People are indeed known to forcefully resist opinions they disagree with, but this resistance is not without bounds and understanding the extent and channels with which contrarian information can lead to attitude change will be important. Another type of explanation revolves around the idea of personality maturation in developmental psychology wherein individuals are assumed to become more agreeable as they age (and hence more receptive to others' arguments and potentially less extreme in their opinions[50]). Finally, a focus on the between-individual heterogeneity in the reasons why people become extremists in the first place might be fruitful in explaining moderation over time. Potentially, extreme attitudes may be driven both by reasons more compatible with transience (such as idiosyncratic life events or social signaling[14,16]) and with resistance to change (such as moralization of policy attitudes).

Finally, the present findings raise the question of when and why societies will tend towards moderation or extremism. Our research was not designed to answer this question, but we can make a few comments. Recent evidence on polarization has focused mainly on affective polarization, namely dislike among political adversaries. However, empirical studies that examined different types of polarization over time showed that not all types of polarization increased in the past decades[3].

Moreover, public interest in disputed socio-political issues ebb and flow over long periods of time. The process of identifying and ultimately resolving issues of conflict can explain how current extreme attitude holders become more moderate over time while new extremists pop up with respect to newly identified issues. Moreover, as new generations join society new crops of potential extremists and moderates enter into the political debate. It is possible that new entrants into the political discourse are more likely to be extreme than people who have been interested and informed about politics for many

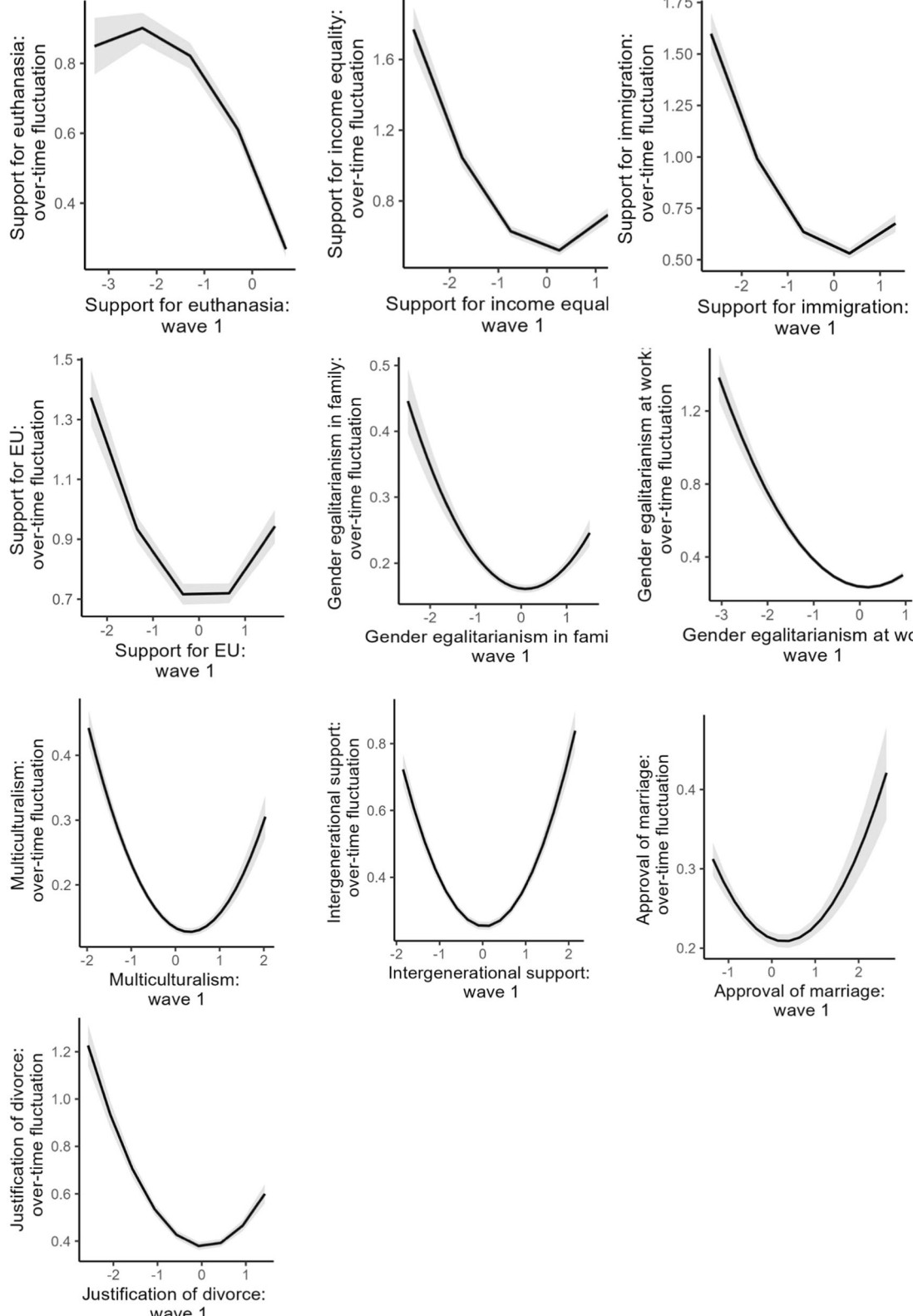

**Fig. 4 Study 2, Holdout sample: Association between attitude at wave 1 and attitude change over 13 waves (measured as within-person standard deviation).** Attitudes at wave 1 were mean-centered; over-time fluctuation is represented MSSD (higher values reflect more fluctuations, 0 reflects complete stability). N = 9310.

years. This would suggest that while (some of) the people who used to be extreme tend to moderate over time, new entrants can nevertheless create polarization.

**Limitations**. This work is not without limitations. What we find differs from previous studies, which measured attitude change over periods of weeks[30] Given that the time horizon appears to

**Table 5 Quadratic moderation analyses, Study 2.**

| Attitude | | Exploratory sample | | | Holdout sample | | |
|---|---|---|---|---|---|---|---|
| | | b | 95% CI | p | b | 95% CI | p |
| Justification of euthanasia | Time linear | 0.01 | 0.01–0.01 | < 0.001 | 0.01 | 0.01–0.01 | < 0.001 |
| | Time quadratic | −0.00 | −0.00 to 0.00 | 0.265 | −0.00 | −0.00 to −0.00 | < 0.001 |
| | Baseline attitude | 0.69 | 0.67–0.71 | < 0.001 | 0.72 | 0.70–0.73 | < 0.001 |
| | Time linear x baseline attitude | −0.02 | −0.02 to −0.02 | < 0.001 | −0.02 | −0.02 to −0.02 | < 0.001 |
| | Time quadratic x baseline attitude | 0.00 | 0.00–0.00 | < 0.001 | 0.00 | 0.00–0.00 | < 0.001 |
| Support for income equality | Time linear | 0.01 | 0.00–0.01 | < 0.001 | 0.01 | 0.01–0.01 | < 0.001 |
| | Time quadratic | −0.00 | −0.00 to 0.00 | 0.621 | −0.00 | −0.00 to 0.00 | 0.932 |
| | Baseline attitude | 0.60 | 0.57–0.62 | < 0.001 | 0.57 | 0.55–0.58 | < 0.001 |
| | Time linear x baseline attitude | −0.03 | −0.03 to −0.03 | < 0.001 | −0.03 | −0.03 to −0.03 | < 0.001 |
| | Time quadratic x baseline attitude | 0.00 | 0.00–0.00 | < 0.001 | 0.00 | 0.00–0.00 | < 0.001 |
| Support for acculturation | Time linear | −0.01 | −0.01 to −0.00 | < 0.001 | −0.01 | −0.01 to −0.01 | < 0.001 |
| | Time quadratic | −0.00 | −0.00 to −0.00 | < 0.001 | −0.00 | −0.00 to −0.00 | < 0.001 |
| | Baseline attitude | 0.61 | 0.59–0.64 | < 0.001 | 0.61 | 0.60–0.62 | < 0.001 |
| | Time linear x baseline attitude | −0.03 | −0.03 to −0.03 | < 0.001 | −0.03 | −0.03 to −0.03 | < 0.001 |
| | Time quadratic x baseline attitude | 0.00 | 0.00–0.00 | < 0.001 | 0.00 | 0.00–0.00 | < 0.001 |
| Disapproval of EU unification | Time linear | 0.01 | 0.01–0.01 | < 0.001 | 0.01 | 0.01–0.01 | < 0.001 |
| | Time quadratic | −0.01 | −0.01 to −0.01 | < 0.001 | −0.01 | −0.01 to −0.01 | < 0.001 |
| | Baseline attitude | 0.58 | 0.56–0.61 | < 0.001 | 0.59 | 0.58–0.60 | < 0.001 |
| | Time linear x baseline attitude | −0.02 | −0.03 to −0.02 | < 0.001 | −0.02 | −0.02 to −0.02 | < 0.001 |
| | Time quadratic x baseline attitude | 0.00 | 0.00–0.00 | < 0.001 | 0.00 | 0.00–0.00 | < 0.001 |
| Gender egalitarianism in family | Time linear | 0.02 | 0.01–0.02 | < 0.001 | 0.01 | 0.01–0.01 | < 0.001 |
| | Time quadratic | 0.00 | −0.00 to 0.00 | 0.131 | −0.00 | −0.00 to 0.00 | 0.586 |
| | Baseline attitude | 0.69 | 0.67–0.71 | < 0.001 | 0.68 | 0.67–0.69 | < 0.001 |
| | Time linear x baseline attitude | −0.02 | −0.02 to −0.02 | < 0.001 | −0.02 | −0.02 to −0.02 | < 0.001 |
| | Time quadratic x baseline attitude | 0.00 | 0.00–0.00 | < 0.001 | 0.00 | 0.00–0.00 | < 0.001 |
| Gender egalitarianism at work | Time linear | 0.01 | 0.01–0.01 | < 0.001 | 0.01 | 0.01–0.01 | < 0.001 |
| | Time quadratic | 0.00 | 0.00–0.00 | < 0.001 | 0.00 | 0.00–0.00 | < 0.001 |
| | Baseline attitude | 0.61 | 0.58–0.64 | < 0.001 | 0.63 | 0.62–0.64 | < 0.001 |
| | Time linear x baseline attitude | −0.02 | −0.02 to −0.02 | < 0.001 | −0.02 | −0.02 to −0.02 | < 0.001 |
| | Time quadratic x baseline attitude | 0.00 | 0.00–0.00 | < 0.001 | 0.00 | 0.00–0.00 | < 0.001 |
| Multiculturalism | Time linear | 0.00 | 0.00–0.01 | < 0.001 | 0.00 | 0.00–0.00 | < 0.001 |
| | Time quadratic | 0.00 | 0.00–0.00 | < 0.001 | 0.00 | 0.00–0.00 | < 0.001 |
| | Baseline attitude | 0.78 | 0.76–0.80 | < 0.001 | 0.77 | 0.76–0.79 | < 0.001 |
| | Time linear x baseline attitude | −0.01 | −0.02 to −0.01 | < 0.001 | −0.01 | −0.02 to −0.01 | < 0.001 |
| | Time quadratic x baseline attitude | 0.00 | 0.00–0.00 | < 0.001 | 0.00 | 0.00–0.00 | < 0.001 |
| Approval of intergenerational support | Time linear | −0.01 | −0.01 to −0.00 | < 0.001 | −0.01 | −0.01 to −0.01 | < 0.001 |
| | Time quadratic | 0.00 | 0.00–0.00 | 0.022 | 0.00 | 0.00–0.00 | < 0.001 |
| | Baseline attitude | 0.63 | 0.60 – 0.66 | < 0.001 | 0.61 | 0.60–0.63 | < 0.001 |
| | Time linear x baseline attitude | −0.02 | −0.02 to −0.02 | < 0.001 | −0.02 | −0.02 to −0.02 | < 0.001 |
| | Time quadratic x baseline attitude | 0.00 | 0.00–0.00 | < 0.001 | 0.00 | 0.00–0.00 | < 0.001 |
| Approval of marriage | Time linear | −0.02 | −0.02 to −0.01 | < 0.001 | −0.02 | −0.02 to −0.01 | < 0.001 |
| | Time quadratic | −0.00 | −0.00 to −0.00 | 0.002 | −0.00 | −0.00 to −0.00 | < 0.001 |
| | Baseline attitude | 0.69 | 0.67–0.71 | < 0.001 | 0.69 | 0.68–0.70 | < 0.001 |
| | Time linear x baseline attitude | −0.02 | −0.02 to −0.02 | < 0.001 | −0.02 | −0.02 to −0.02 | < 0.001 |
| | Time quadratic x baseline attitude | 0.00 | 0.00–0.00 | < 0.001 | 0.00 | 0.00–0.00 | < 0.001 |
| Justification of divorce | Time linear | 0.01 | 0.01–0.01 | < 0.001 | 0.01 | 0.01–0.01 | < 0.001 |
| | Time quadratic | 0.00 | −0.00 to 0.00 | 0.085 | 0.00 | 0.00–0.00 | < 0.001 |
| | Baseline attitude | 0.55 | 0.52–0.57 | < 0.001 | 0.57 | 0.56–0.59 | < 0.001 |
| | Time linear x baseline attitude | −0.03 | −0.03 to −0.02 | < 0.001 | −0.02 | −0.03 to −0.02 | < 0.001 |
| | Time quadratic x baseline attitude | 0.00 | 0.00–0.00 | < 0.001 | 0.00 | 0.00–0.00 | < 0.001 |

All predictors were centered; all models included a random intercept at the level of participants. $N = 11{,}570$.

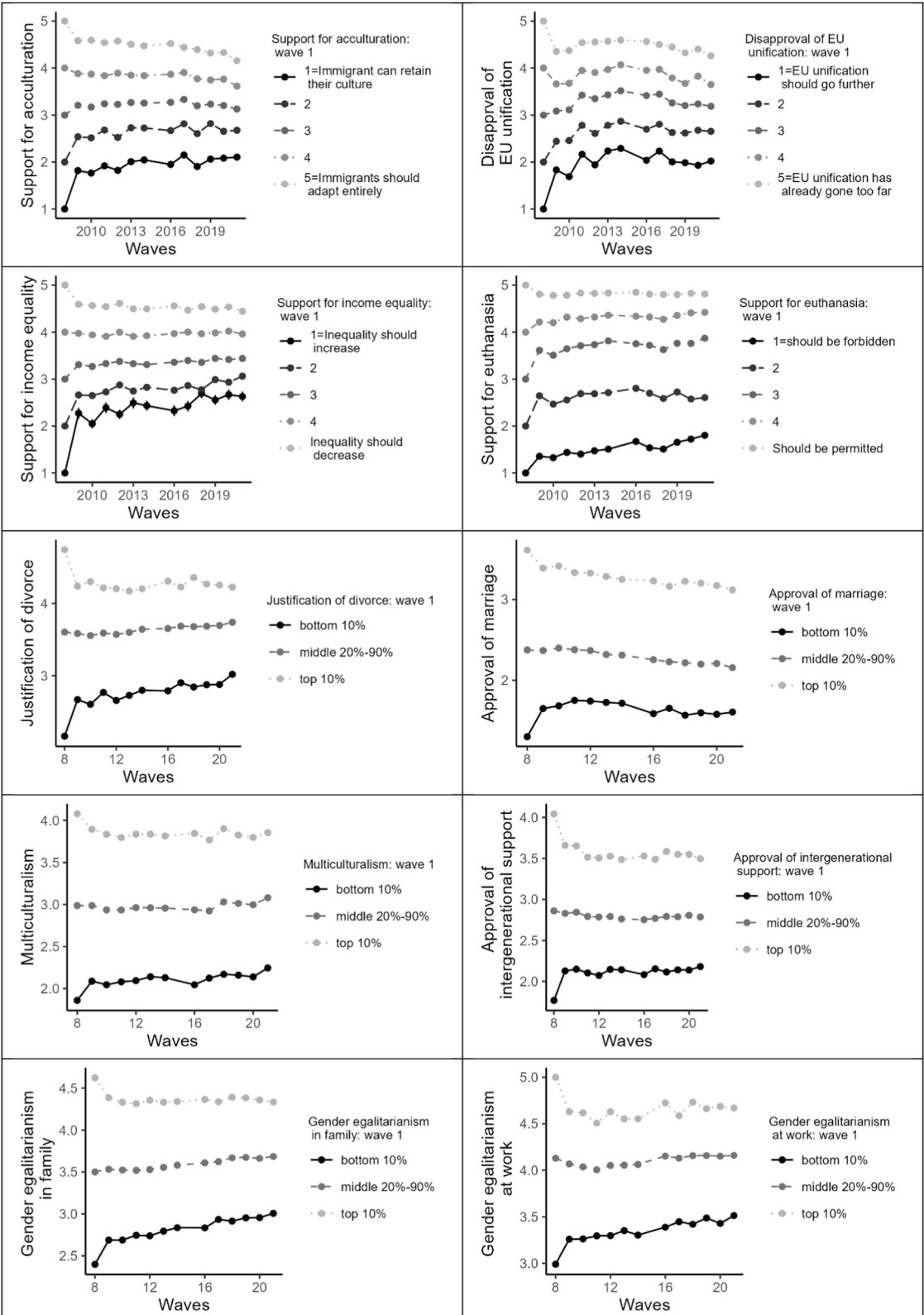

**Fig. 5 Study 2, Holdout sample: Attitude change trajectories as a function of initial attitude (LISS Panel).** Attitudes measured with single items: developmental trajectory associated with each response category is shown; attitudes measured with multi-item scales: developmental trajectories of bottom 10%, middle 20–90%, and top 10% are shown. Each point includes error bars that represent standard errors (too small to be discerned). N = 9310.

matter, it would be especially helpful to measure attitudes over even longer time periods, such as decades in order to understand the long-run evolution of extreme and moderate attitude holders.

Second, many socio-political issues experience fluctuation sin the level of the interest they evoke among partisans in different time periods. Our study does not explicitly account for this or for

the reasons why some issues become more or less central. It would be very helpful to understand how the general interest in an issue affects the evolution of extreme attitudes related to it.

Third, our studies focus on Western samples, namely from the Netherlands and the United States. Due to various cultural and political differences, it would be very helpful to understand samples from Eastern cultures as well.

Fourth, our research provides evidence for effects but does not provide evidence for psychological mechanisms underlying these effects. More than mechanism can explain effects such as these, which occur over long periods of time and across very different socio-political issues. Although we raise ideas for these mechanisms, future research can test them.

**Concluding thought**. The present findings suggest that for policy attitudes, over multi-year periods, and at the individual level extremists tend to change more than moderates, and when extremists do change the direction is towards moderation. The full picture of the state of political disagreement may thus be more complex than is often assumed. Yet, our finding that extremists can moderate over time offer hopeful assessments of the prospects for conflict mitigation.

## Data availability
Data are available at https://osf.io/57gpj/?view_only=53b137e2f7974699ac7f40c6c3a8ec04.

## Code availability
Code is available at https://osf.io/57gpj/?view_only=53b137e2f7974699ac7f40c6c3a8ec04.

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

## Acknowledgements
We thank the Hoffmann Global Institute for Business and Society at INSEAD for funding. The funders had no role in study design, data collection and analysis, decision to publish or preparation of the manuscript.

## Author contributions
N.K. formulated hypotheses, conducted Study 3, and wrote the manuscript. O.S. formulated hypotheses, analyzed the data for Studies 1–2, and provided revisions to the manuscript.

## Competing interests
The authors declare no competing interests.
