## [Peer Review File · Communications Psychology]

29th Jun 23

Dear Professor Klein,

Thank you for your patience during the peer-review process. Your manuscript titled "The evolution of political extremists: In policy and ideology, extreme attitude holders tend to converge towards the middle over time" has now been seen by 3 reviewers, whose comments are appended below. You will see that they find your work of some potential interest. However, they have raised quite substantial concerns that must be addressed. In light of these comments, we cannot accept the manuscript for publication, but would be interested in considering a revised version that fully addresses these serious concerns.

We hope you will find the Reviewers' comments useful as you decide how to proceed. Should additional work allow you to address these criticisms, we would be happy to look at a substantially revised manuscript. If you choose to take up this option, please highlight all changes in the manuscript text file, and provide a detailed point-by-point reply to the reviewers.

Editorially, we prioritize the following issues.

Reviewer #2 and #3 mention a range of concerns that need to be addressed through additional analysis. The most pressing matter is the concern that the results may be a statistical artefact of regression to the mean, or bounded scales. You need to provide suitable additional analysis to alleviate these concerns. In a similar vein, in your first preregistration, you note that regression to the mean may be the driver of the expected effect - please comment on whether you were referring to an actual behaviour change (towards more moderate views) or a measuring effect.

Following the inclusion of suitable additional analyses, please ensure that any remaining ambiguities are discussed transparently in a section titled "Limitations" in the Discussion. Because your study was preregistered, please ensure that all preregistered analyses are included and labelled as such. Exploratory analyses should likewise be labelled accordingly.

The referee reports highlight that some analysis choices require better justification, please strengthen the rationale you provide for these (or implement suitable alternative analyses).

Finally, alternative mechanisms should also be discussed in the Discussion (though not necessarily as a limitation).

For the experiment reported in the Supplementary Information, please report whether ethics approval or an IRB waiver were in place and whether participants provided informed consent. We recommend moving the description of the experiment and its results to the main manuscript.

If the revision process takes significantly longer than five months, we will be happy to reconsider your paper at a later date, provided it still presents a significant contribution to the literature at that stage.

Please use the following link to submit your revised manuscript, point-by-point response to the Reviewers' comments with a list of your changes to the manuscript text (which should be in a separate document to any cover letter) and any completed checklist:

[link redacted]

Please do not hesitate to contact me if you have any questions or would like to discuss the required revisions further. Thank you for the opportunity to review your work.

Best regards,

Yafeng Pan

Yafeng Pan, PhD
Editorial Board Member
Communications Psychology
orcid.org/0000-0002-5633-8313

EDITORIAL POLICIES AND FORMATTING

Editorial Policy: Policy requirements (Download the link to your computer as a PDF.)

Furthermore, please align your manuscript with our format requirements, which are summarized on the following checklist:

Communications Psychology formatting checklist

and also in our style and formatting guide Communications Psychology formatting guide .

* **CODE AVAILABILITY:** All Communications Psychology manuscripts must include a section titled "Code Availability" at the end of the methods section. In the event of publication, we require that the custom analysis code supporting your conclusions is made available in a publicly accessible repository; please choose a repository that provides a DOI for the code; the link to the repository and the DOI must be included in the Code Availability statement. Publication as Supplementary Information will not suffice. We ask you to prepare and upload code at this stage, to avoid delays later on in the process.

* **DATA AVAILABILITY:**

All Communications Psychology research manuscripts must include a section titled "Data Availability" at the end of the Methods section or main text (if no Methods). More information on this policy, is available at http://www.nature.com/authors/policies/data/data-availability-statements-data-citations.pdf.

At a minimum the Data availability statement must explain how the data can be obtained and whether there are any restrictions on data sharing. Communications Psychology strongly endorses open sharing of data. If you do make your data openly available, please include in the statement:

We recommend submitting the data to discipline-specific, community-recognized repositories, where possible and a list of recommended repositories is provided at http://www.nature.com/sdata/policies/repositories.

If a community resource is unavailable, data can be submitted to generalist repositories such as figshare or Dryad Digital Repository. Please provide a unique identifier for the data (for example a DOI or a permanent URL) in the data availability statement, if possible. If the repository does not provide identifiers, we encourage authors to supply the search terms that will return the data. For data that have been obtained from publicly available sources, please provide a URL and the specific data product name in the data availability statement. Data with a DOI should be further cited in the methods reference section.

Please refer to our data policies at http://www.nature.com/authors/policies/availability.html.

REVIEWER REPORTS:

Reviewer #1 (Remarks to the Author):

This paper provides a compelling analysis of the relative likelihood of changes in views over time by extremists and moderates (as measured on survey scales). In studies utilizing data from the US and Denmark, over time measures of attitudes about an array of social issues found that over time those who started the series at the extreme tended (a) to evidence more variation in expressed attitude over time than did moderates and (b) those initially on the extremes tended to moderate toward the middle. This is contrary to the theoretical (and casual) conjecture that extremists will hold the most intractable and rigid attitudes over time.

The methods used in the paper are appropriate, and well documented. I particularly appreciated the careful visualizations used to characterize the results.

The findings are consistent across the two data sets, and the implications are important. Loosening the expectation that those with extreme attitudes won't change their minds would seem to make strategies of continuing engagement more efficacious, and would reduce the sense that democratic processes are trapped by entrenched polarized beliefs.

The paper needs to be edited carefully. One error (at least I think it's an error) reverses the main findings. In discussing the results of study 2, the authors say: "For all attitudes but one (euthanasia), we detected a pattern where extremists exhibited less attitude fluctuations over time than moderates." I think they meant to say that extremists exhibited more attitude fluctuations. This and other minor errors undermine the important message of the paper.

Overall I believe this is an important paper, and after careful editing it should be published.

Reviewer #2 (Remarks to the Author):

Using the General Social Survey and the Longitudinal Internet Studies for the Social Sciences, this paper examines whether people with extreme values are more likely to change their attitudes compared to those with moderate values. Although contradicting many previous empirical studies on attitude and cultural change, their results are consistent through the two datasets used in the survey. I think this is an innovative work, and I have a few questions about the manuscript.

1. The authors listed reasons that shows it might not be the case that people with extreme values can change. But it's not enough reason for them to change more than the moderates and focus on

individual attitude change. They didn't specify the mechanism for systematic changes for people holding more extreme values. Kiley and Vaisey (2020) listed some works that suggest the mechanism of how social environments influence individual attitudes and behaviors. It's possible that individuals with extreme values have more commitment to their cultural dispositions and adopt behaviors and values that are consistent with their cultural dispositions (Goldberg and Stein 2018). Or that they adapt their partisan affiliations or ideological commitments to conform to their social groupings (Baldassarri and Gelman 2008). Although this paper does not explore the reason for extremists' attitude change, it is still necessary to motivate the analysis with more social theories.

2. The overall result shows patterns of extremists changing more than moderates. It's not a usual way to frame this pattern but it's not wrong. Thus it requires more explanation of possible mechanisms in the paper.

3. Is this paper capturing more than regression to the mean? I appreciate the treatment of the question in the discussion, as the authors try to get ahead of it. That alone is encouraging, regardless of whether the argument is convincing. It could also be addressed with corroborative analyses. The subset of people whose identity changes from moderate to extreme, what is the ratio of their change in policy attitude to their change in identify? That ratio should be larger than the same ratio for the subset whose identity went from extreme to moderate. This or some other kind of extra supporting analysis is definitely called for. The authors describe their own supplementary analysis, and what they describe is a good internal self-replication, but I'm not satisfied that it's a control for regression to the mean. Unless I am misunderstanding, the study two analysis is just as vulnerable as the first to the problem it is supposed to address.

4. The definition of extreme attitude holders needs to be defined better in the paper. In the analysis, extremity is only defined at the level of single measures instead of multiple dimensions of beliefs or behaviors, which are more commonly used in other studies.

5. GSS collected information on a wide range of attitudinal questions and the authors selected a list of measures. This selection covers many important socio-political issues but is not an exhaustive list. What's the standard for this selection of measures?

6. In the analysis to test whether extremists are less likely to change policy attitudes than moderates, do the authors include a trend variable? Because attitude changes in social changes are mainly related to changes in social environments, and the authors did combine the three time periods, it is crucial to consider the difference in social environments in the model.

7. The regression used to estimate whether extremists change attitudes in a different trajectory from moderates(Q2) needs to be better justified in Study 1, especially the interpretation of the results. I have a few questions or concerns about this analysis. First, a quadratic time term with three timepoints is unusual. I'm also a bit concerned with how much the interpretation relies on an interaction term (b5). Last, I don't think this analysis can address the author's second question: whether the direction of change in extremists (vs. moderates) tends towards the extreme or the middle. Figure 2 seems to better address this question.

8. Figure 1 results shows that extremist had more change than moderates. Is this due to the change in variance in the overall distribution? It would be helpful to see the evolution of overall distribution.

9. The authors mention that this analysis can help us understand whether the polarization is due to extremists being stubborn. Are the items the authors selected already polarized? If so, are they normally distributed or bimodal? If bimodal, do estimations based on mean still work?

10. The authors did an additional analysis of ideology to explain the attitude change of extreme and moderate values. We see the same regression toward the mean in the ideology, but how this result can explain the differential attitude change on policies is still unclear in the paper.

11. In addition to “stubborn extremist” theory, there’s the potential competing predictions of the “attention-seeking extremist” who we would expect to be dynamic in their policy attitudes, and probably in a direction that is anti-correlated with others. I’m not proposing that “attention-seeking extremist” is a current academic theory of extremism (I don’t know either way) but it’s at least enough of an established folk theory to merit consideration, especially if it has a chance of helping to motivate the paper.

Reference

Baldassarri, D., & Gelman, A. (2008). Partisans without constraint: Political polarization and trends in American public opinion. *American Journal of Sociology*, 114(2), 408-446.

Goldberg, A., & Stein, S. K. (2018). Beyond social contagion: Associative diffusion and the emergence of cultural variation. *American Sociological Review*, 83(5), 897-932.

Kiley, K., & Vaisey, S. (2020). Measuring stability and change in personal culture using panel data. *American Sociological Review*, 85(3), 477-506.

Recommend major revision

Reviewer #3 (Remarks to the Author):

This paper tests two hypotheses: those with extreme views change less than those with moderate views; and those with extreme views moderate over time. Using evidence from the GSS and the Dutch LISS panel, the authors find strong support for both hypotheses.

While this paper tackles a fascinating questions, I believe that both tests suffer from important shortcomings that are so far insufficiently addressed in the manuscript. These are as follows:

- Moderation over time: My main concern here is that attitude scales are inherently bounded. On a 1-5 scale, someone who answers 1 can only be recorded in the next wave as staying stable or moderating, even if their views in fact become more extreme. Thus, there is an inherent underestimation of attitude change towards more extreme points. Hence, the finding may be an artefact of response behaviour rather than a true change in attitudes. This challenge is somewhat addressed by multi-item scales, but of course it does not disappear fully.

- Implications: An implication of the moderation over time finding is that, as time goes on, societies everywhere should tend towards moderation. If extremists almost always become more moderate at a higher rate than moderates become extreme, then why do we ever witness polarization? How can this finding be squared with this? This increases my concern that this finding is a methodological artifact.

- Levels of change: the other key finding is that extremists change more than moderates. Again, I have an important methodological concern, namely that many people do not hold issue attitudes at all. Satisficing is a common type of behaviour on surveys. Those without an attitude may often choose midpoint positions, and may choose those consistently over time. If we compared those who truly hold an attitude, would the findings be the same? I am not sure. At the very least, it would be important to take political interest, engagement, levels of education or cognitive abilities into account in the analysis.

- Differential dropout rates: Do the authors have information on whether those with extreme views are more likely to drop out in between waves? This is another alternative explanation.

Smaller points:

- Graphs should always include confidence intervals, yet none of the graphs with quadratic effects do.

- I am not convinced by imposing a quadratic association on all regressions. I would prefer to (also) see regressions with indicator variables for different levels. Imposing quadratic terms can lead to unusual consequences for predicted levels.

1

Below we provide responses to reviewers.

Reviewer #1:

R1 writes: **“This paper provides a compelling analysis of the relative likelihood of changes in views over time by extremists and moderates (as measured on survey scales). In studies utilizing data from the US and Denmark, over time measures of attitudes about an array of social issues found that over time those who started the series at the extreme tended (a) to evidence more variation in expressed attitude over time than did moderates and (b) those initially on the extremes tended to moderate toward the middle. This is contrary to the theoretical (and casual) conjecture that extremists will hold the most intractable and rigid attitudes over time.**

The methods used in the paper are appropriate, and well documented. I particularly appreciated the careful visualizations used to characterize the results.

The findings are consistent across the two data sets, and the implications are important. Loosening the expectation that those with extreme attitudes won’t change their minds would seem to make strategies of continuing engagement more efficacious, and would reduce the sense that democratic processes are trapped by entrenched polarized beliefs.

We are grateful for the careful review and these helpful and encouraging comments.

The paper needs to be edited carefully. One error (at least I think it’s an error) reverses the main findings. In discussing the results of study 2, the authors say: “For all attitudes but one (euthanasia), we detected a pattern where extremists exhibited less attitude fluctuations over time than moderates.” I think they meant to say that extremists exhibited more attitude fluctuations. This and other minor errors undermine the important message of the paper.

Thank you very much for noticing this error – we are grateful and have corrected it. Further, we have gone through the manuscript carefully in our best attempt to avoid similarly careless errors.

Overall I believe this is an important paper, and after careful editing it should be published.”

As before, we note our appreciation and gratitude for the very helpful comments.

Reviewer #2:

R2 writes: **“Using the General Social Survey and the Longitudinal Internet Studies for the Social Sciences, this paper examines whether people with extreme values are more likely to change their attitudes compared to those with moderate values. Although contradicting many previous empirical studies on attitude and cultural change, their results are consistent through the two datasets used in the survey. I think this is an innovative work, and I have a few questions about the manuscript.**

1. The authors listed reasons that shows it might not be the case that people with extreme values can change. But it’s not enough reason for them to change more than the moderates and focus on individual attitude change. They didn’t specify the mechanism for systematic changes for people holding more extreme values. Kiley and Vaisey (2020) listed some works that suggest the mechanism of how social environments influence individual attitudes and

behaviors. It's possible that individuals with extreme values have more commitment to their cultural dispositions and adopt behaviors and values that are consistent with their cultural dispositions (Goldberg and Stein 2018). Or that they adapt their partisan affiliations or ideological commitments to conform to their social groupings (Baldassarri and Gelman 2008). Although this paper does not explore the reason for extremists' attitude change, it is still necessary to motivate the analysis with more social theories.

We respond to this and the following point jointly (after your comment 2).

2. The overall result shows patterns of extremists changing more than moderates. It's not a usual way to frame this pattern but it's not wrong. Thus it requires more explanation of possible mechanisms in the paper.

We revised and clarified the possible mechanisms we believe might help explain why extreme attitude holders might change more than moderates (pp. 4-5). In doing so, we incorporate some of the citations you provided – thank you.

First, temporary situations (i.e. becoming a first-time parent; Eibach et al., 2003) can make people more extreme in various issues (becoming “tough on crime”). When these situations are over, people may well moderate their attitudes. In the manuscript, we provide multiple examples of findings showing that people becoming more moderate due to temporary life situations only to return to extreme attitudes when those situations run their course. Thus, this potential mechanism is can lead extreme attitude holders becoming more moderate.

Second, existing research suggests that mechanistically explaining policy issues can moderate attitudes (Fernbach et al., 2013). It is possible that learning more about issues over time can lead extreme attitude holders to moderate their positions. This possibility is consistent with work suggesting that information acquisition can lead to attitude change (Page & Shapiro, 1992) and that attitude change tends to occur more frequently among younger people (Kiley & Vaisey, 2020) who may be less knowledgeable and less capable of explaining nuances of policy issues compared to older people. In contrast, to our knowledge no research suggests that more knowledge about how policies work can lead moderates to become more extreme.

Third, it is possible the extreme attitude holders identify and follow their chosen political parties more strongly than moderates (Baldassari & Gelman, 2008). This in turn may mean that extreme attitude holders would follow the party line even when their chosen political party makes compromises as a way of expanding its electoral reach.

Finally (based on Reviewer 2's idea), people might adopt extreme attitudes to stand out from the mainstream. Indeed, political extremists tend to score lower on conformity values (Rigoli, 2023). Since what is considered “mainstream” tends to change over time, extremists might abandon their extreme position on the issues where they anticipate their (originally extreme) views to become more conventional, common, and widely-endorsed in the near future (the same way early adopters abandon fashion items that become mainstream).

Eibach, R. P., Libby, L. K., & Gilovich, T. (2003). When a change in the self is mistaken for a change in the world. *Journal of Personality and Social Psychology*, 84, 917-931.

Fernbach, P. M., Rogers, T., Fox, C. R., & Sloman, S. A. (2013). Political extremism is supported by an illusion of understanding. *Psychological Science*, 24, 939-946.

Rigoli, F. (2023). Political Extremism and a Generalized Propensity to Discriminate Among Values. *Political Psychology*, 44, 301-318.

R2 writes: **“3. Is this paper capturing more than regression to the mean? I appreciate the treatment of the question in the discussion, as the authors try to get ahead of it. That alone is encouraging, regardless of whether the argument is convincing. It could also be addressed with corroborative analyses. The subset of people whose identity changes from moderate to extreme, what is the ratio of their change in policy attitude to their change in identify? That ratio should be larger than the same ratio for the subset whose identity went from extreme to moderate. This or some other kind of extra supporting analysis is definitely called for. The authors describe their own supplementary analysis, and what they describe is a good internal self-replication, but I’m not satisfied that it’s a control for regression to the mean. Unless I am misunderstanding, the study two analysis is just as vulnerable as the first to the problem it is supposed to address.**

[Follow-up from subsequent email exchange with R2 copied below]

The key issue is that if there is error in the measurement of attitudes, then for spurious reasons an agent measuring as an extreme attitude holder in one time step (whether step 1 or step 2) is more likely to measure as moderate in the next step, presumably because they were always moderate and were just measured under high noise. The reviewer is concerned that the solution you offer for this concern, to self-replicate from step 2 data is not in fact a control for regression to the mean, because it is equally vulnerable to regression to the mean. To alleviate this concern, detecting a drift toward moderate views above and beyond mere regression to the mean requires a bit more. For this, you may benefit from the additional data you gathered on both attitudes and other info from the General Social Survey (including political identification?). If these vary independently, and/or in equal degrees in both directions (from extreme to moderate attitudes, and moderate to extreme) then the spurious/statistical explanation in terms of regression to the mean seems hard to dismiss. If however, there is a more theoretical explanation for these observations, particularly that suggested by you in the manuscript already, we would expect a change in attitude to be attended by a change in overall political identity (or some proxy for it). Put another way, if these effects are real then we should expect various survey indicators of extreme views to change together from extreme to moderate, and likely not the other way to do it.

The particular measure included in the previous review, differences in the ratios of attitudes to some ideology, is just one way of getting at this broader issue, that we can disqualify or weaken explanations in terms of statistical artifacts if, within individual, several different measures go together from extreme to moderate, particularly if attitude changes in the other direction, from moderate to extreme, do not have the same within-subject coherence.

We thank R2 for taking the time to elaborate more on this important issue. We agree that focusing on the patterns of change among different attitude items over time is very worthwhile. We developed it further and think that the new additional analyses help address concerns related to regression to the mean. We elaborate on this approach below.

Following the reviewer’s suggestion, we too believe that if several attitude items show a similar pattern of change within participants, that would substantiate the interpretation of the results as reflecting an actual change rather than regression to the mean/measurement error. However, we would like to add that this can only be the case for items that reflect similar attitudinal constructs and are thus correlated with each other from the beginning. If A and B reflect related / similar constructs, it’s logical to expect that they will change in a similar way over time. On the other hand,

if correlated change is present in any two randomly selected items (including ones that are not semantically related to each other and thus measure unrelated attitudinal constructs), observing correlated change would likely reflect a statistical artefact (e.g., regression to the mean).

Fortunately, our datasets included a large set of attitude items, some of which were conceptually related to each other while others were not. In Study 1, correlations (at wave 1) ranged from $r = -.42$ (political ideology and support of gays/lesbians) to $r = .56$ (permissive sex attitudes and support of gays/lesbians). In an additional set of analyses, for each individual and each attitude, we computed a **change score** by subtracting attitude at the last wave from the attitude value at the first wave ($[\text{attitude}_{\text{first}} - \text{attitude}_{\text{last}}]$). Higher values in this change score indicate that a participant's value on a specific attitude item decreased over time and lower values indicate that it increased over time.

We then computed **the correlated change coefficient for each pair of attitude items**. They ranged from $r = -.11$ (political ideology and support of gays/lesbians; participants who experienced an "increase" in political ideology [i.e., became more conservative] also decreased their support of gays/lesbians) to $r = .20$ (permissive sex attitudes and support of gays/lesbians; participants who experienced an increase in permissive sex attitudes also experienced an increase in support of gays/lesbians). This analysis shows that some attitude items exhibited a coordinated / correlated change while others did not.

Next we tested whether conceptually similar attitudes (i.e., attitudes that were correlated with each other at wave 1) showed a similar pattern of change. Again, our assumption is that if attitude items that are conceptually similar (i.e., are correlated with each other) show a similar pattern of change, that is likely to indicate actual attitude change (rather than a statistical artefact). In contrast, if attitude items that are not conceptually similar (i.e., are not correlated with each other) show similar pattern of change, that would reflect a statistical artefact (e.g., regression to the mean). The association between correlated change and initial attitude similarity is plotted below (Study 1: $n = 72$ issue pairs, $r = .82$, $p < .001$). It shows that similar constructs had similar change trajectories. On the other hand, correlated change did not emerge for pairs of issues that measured unrelated attitudinal constructs, suggesting that the observed correlated change is unlikely to reflect statistical artefact (e.g., regression to the mean). The same pattern emerged in Study 2 (Study 2: $n = 110$ issue pairs, $r = .91$, $p < .001$).

Note. Each point represents a pair of issues (e.g., permissive sex attitudes and support of gays/lesbians); the x-axis shows how similar issues in each pair are at baseline (i.e., do people who scored high on one issue scored high on the other one?) and the y-axis shows how similarly issues in each pair changed over time (i.e., do people who increased on one issue increased on the other one as well?).

R2 writes: **“4. The definition of extreme attitude holders needs to be defined better in the paper. In the analysis, extremity is only defined at the level of single measures instead of multiple dimensions of beliefs or behaviors, which are more commonly used in other studies.”**

Our analyses included attitudes towards multiple different policy issues, from economy to family to gender issues. Consequently, these various attitudes items were not sufficiently strongly related to each other to justify combining them into a single dimension. We show these correlations in Tables 7S and 8S in the SOM.

More importantly, on pp. 35-39 we provide an analysis of extremists and moderates on the dimension of general ideology in the GSS (American participants) and the LISS (Dutch participants). General ideology is a more complex attitude that likely encapsulates various granular attitudes that coalesce into being “right wing” or “left wing.” We find similar results with general ideology as we find with attitudes about separate socio-political issues. This, we believe, gives confidence that the results did not emerge due to defining attitude extremity on the level of single issues.

R2 writes: **“5. GSS collected information on a wide range of attitudinal questions and the authors selected a lots of measures. This selection covers many important socio-political issues but is not an exhaustive list. What’s the standard for this selection of measures?”**

The criteria for the selection of measures were specified in the pre-registration and were determined

based on the practical necessity of being able to measure extremeness over time: “The dataset includes multiple questions assessing participants’ attitudes towards different socio-political issues. Our analyses will include all attitude items on socio-political issues that were measured in all three waves. As we need to measure attitude extremity, we will only select the attitude items that were using a continuous scale allowing us to assess extreme vs. moderate support / opposition to an issue.” - <https://osf.io/h67bk>.

R2 writes: “6. In the analysis to test whether extremists are less likely to change policy attitudes than moderates, do the authors include a trend variable? Because attitude changes in social changes are mainly related to changes in social environments, and the authors did combine the three time periods, it is crucial to consider the difference in social environments in the model.”

The first set of analyses that predicts over-time fluctuation in attitudes, which uses the linear and the quadratic terms of baseline attitudes, cannot include a trend variable because this analysis is conducted at the individual level (each row of data represents an individual; including a trend variable is possible only for longitudinal analyses, where each row of data represents an individual’s response in each wave).

However, the trend variable is included in the second set of analyses where we model the attitude trajectory over time for each participant as a function of the extremity of the participant’s initial attitude (coefficient b1 in the formula on p. 10 reflects the effect of time – a trend variable).

In case you are referring to the possibility that average responses could have changed across time and this change could have influenced one’s position at the extreme vs. the middle of the distribution, please see our responses to comment 8 below.

R2 writes: “7. The regression used to estimate whether extremists change attitudes in a different trajectory from moderates(Q2) needs to be better justified in Study 1, especially the interpretation of the results. I have a few questions or concerns about this analysis. First, a quadratic time term with three timepoints is unusual. I’m also a bit concerned with how much the interpretation relies on an interaction term (b5). Last, I don’t think this analysis can address the author’s second question: whether the direction of change in extremists (vs. moderates) tends towards the extreme or the middle. Figure 2 seems to better address this question.”

Figure 2 provides an illustration of the results of the quadratic regression analyses. The significance of the interaction term (b5) suggests that individuals with initially higher/lower vs. moderate values show different trajectories over time (Hayes, 2015; cited in the manuscript). Figure 2 simply plots these trajectories.

Nevertheless, to address your concern, we conducted an additional set of analyses in which we treated both the time variable and the attitude variables as categorical. The time variable included three categories representing the three time points. For categorizing the attitude variables, we followed the strategy used in Figure 2 (Study 1, pp. 18-19 in the manuscript) and Figure 5 (Study 2, pp. 32-33 in the manuscript): attitudes measured with multi-item scales were split into three categories: bottom 10%, moderate (middle 20%-90%) and top 10%. The model included time, initial attitudes and the interaction terms among them. The significance of the interaction terms indicate that the time effects are moderated by the initial attitudes.

The resulting coefficients are presented in Table 9S (pp. 21-23 in the SOM) and the simple slope analysis (i.e., the effect of time for individuals who score low, moderately or high at baseline)

results are presented in Table 10S (pp. 24-25 in the SOM). The results support the conclusions from the analyses using quadratic terms. Specifically, one can see that for individuals with extremely low (vs. high) original attitudes, the effect of time is significant and positive (vs. negative), while for individuals with moderate original attitudes, the effect of time is either nonsignificant or is substantially smaller.

R2 writes: **“8. Figure 1 results shows that extremist had more change than moderates. Is this due to the change in variance in the overall distribution? It would be helpful to see the evolution of overall distribution.”**

We plotted the distribution of each attitude for each wave (see Figures 5S and 6S, pp. 26-30 in the SOM).

In addition, we added additional analyses wherein attitude values were standardized within each wave. In this case, each individual value reflects a person's position relative to others in each respective wave. These analyses also showed that individuals who held more extreme attitudes (compared to others) initially, became less extreme (compared to others) in the follow-up (see Tables 7S and 8S, pp. 31-32 in the SOM).

R2 writes: **“9. The authors mention that this analysis can help us understand whether the polarization is due to extremists being stubborn. Are the items the authors selected already polarized? If so, are they normally distributed or bimodal? If bimodal, do estimations based on mean still work?”**

Please see our response to comment 8. Some of the variables show bi-modal distributions. It is important to note that in linear mixed models, there is no formal statistical requirement for the response variable to be normally distributed.

R2 writes: **“10. The authors did an additional analysis of ideology to explain the attitude change of extreme and moderate values. We see the same regression toward the mean in the ideology, but how this result can explain the differential attitude change on policies is still unclear in the paper.”**

We now realize that we did not explain the purpose of the analysis of ideology well. We did not mean it as intending to explain attitude change of extremists vs. moderates, but rather as a robustness test to see whether our results extend beyond specific socio-political issues to general ideologies. We have clarified this on pp. 35.

R2 writes: **“11. In addition to “stubborn extremist” theory, there’s the potential competing predictions of the “attention-seeking extremist” who we would expect to be dynamic in their policy attitudes, and probably in a direction that is anti-correlated with others. I’m not proposing that “attention-seeking extremist” is a current academic theory of extremism (I don’t know either way) but it’s at least enough of an established folk theory to merit consideration, especially if it has a chance of helping to motivate the paper.**

This is an interesting possibility, and we appreciate the intent of helping to motivate the paper. We have added this potential mechanism to the paper (p. 5).

We appreciate the suggestion.

R2 writes: **“References**

Baldassarri, D., & Gelman, A. (2008). Partisans without constraint: Political polarization and trends in American public opinion. *American Journal of Sociology*, 114(2), 408-446.

Goldberg, A., & Stein, S. K. (2018). Beyond social contagion: Associative diffusion and the emergence of cultural variation. *American Sociological Review*, 83(5), 897-932.

Kiley, K., & Vaisey, S. (2020). Measuring stability and change in personal culture using panel data. *American Sociological Review*, 85(3), 477-506.

Recommend major revision.”

We are grateful to R2 for the very helpful comments.

Reviewer #3:

R3 writes: **“This paper tests two hypotheses: those with extreme views change less than those with moderate views; and those with extreme views moderate over time. Using evidence from the GSS and the Dutch LISS panel, the authors find strong support for both hypotheses.**

While this paper tackles a fascinating questions, I believe that both tests suffer from important shortcomings that are so far insufficiently addressed in the manuscript. These are as follows:

- Moderation over time: My main concern here is that attitude scales are inherently bounded. On a 1-5 scale, someone who answers 1 can only be recorded in the next wave as staying stable or moderating, even if their views in fact become more extreme. Thus, there is an inherent underestimation of attitude change towards more extreme points. Hence, the finding may be an artefact of response behaviour rather than a true change in attitudes. This challenge is somewhat addressed by multi-item scales, but of course it does not disappear fully.

This is an astute point. To deal with this issue, we repeated the analyses using Tobit regression. Tobit regression allows obtaining unbiased coefficients in the presence of floor or ceiling effects (McBee, 2010; Wang et al. 2008). The results supported the conclusions of our original analyses, suggesting that bounded scales did not explain our original effects.

We included a summary note about the Tobit analysis in the main manuscript (pp. 21 in the manuscript) and include the complete Tobit analysis in the SOM because the main manuscript is getting quite long. We are happy to move it to the main manuscript if necessary.

Tables 3S-6S (pp. 13-17 in the SOM) present the results of the Tobit analyses.

McBee, M. (2010). Modeling Outcomes with Floor or Ceiling Effects: An Introduction to the Tobit Model. *Gifted Child Quarterly*, 54(4), 314–320. <https://doi.org/10.1177/0016986210379095>

Wang, L., Zhang, Z., McArdle, J. J., & Salthouse, T. A. (2008). Investigating ceiling effects in longitudinal data analysis. *Multivariate behavioral research*, 43(3), 476-496.

R3 writes: “- **Implications: An implication of the moderation over time finding is that, as time goes on, societies everywhere should tend towards moderation. If extremists almost always become more moderate at a higher rate than moderates become extreme, then why do we ever witness polarization? How can this finding be squared with this? This increases my concern that this finding is a methodological artifact.**

We do not believe that our findings suggest that societies will necessarily always get more moderate over time.

First, existing evidence on rising polarization has focused mainly on affective polarization, namely the mutual dislike among political adversaries. This type of polarization differs from polarization of socio-political issues, which we measure here. It is entirely possible for polarization to increase in terms of mutual dislike but decrease in terms of policy issues. In fact, empirical studies that examined different types of polarization over time showed that not all types of polarization increased in the past decades (Lelkes, 2016).

Second, public interest in disputed socio-political issues ebb and flow over long periods of time. For example, the national public debt was a major source of acrimonious contention in the United-States in the 1980s whereas today it is far less so. As another example, gay marriage has become mainstream whereas in the past it had been a lightning rod for conflict. The process of identifying and ultimately resolving issues of conflict can explain how current extremists become more moderate over time while new extremists pop up over time with respect to newly identified issues.

Third, as new generations join society new crops of potential extremists and moderates enter into the political debate. It is possible that new entrants into the political discourse are more likely to be extreme than people who have been interested and informed about politics for years. In other words, the young might on average be more extreme than the old – not an unreasonable possibility. This would suggest that while (some of) the people who used to be extreme tend to moderate over time, new entrants can nevertheless create polarization.

We provide a brief discussion of these points on pp. 48-49. Thank you for bringing this to our attention, as we believe the paper is better as a result of including this discussion.

Lelkes, Y. (2016). Mass Polarization: Manifestations and Measurements, *Public Opinion Quarterly*, 80(S1), 392–410, <https://doi.org/10.1093/poq/nfw005>

R3 writes: “- **Levels of change: the other key finding is that extremists change more than moderates. Again, I have an important methodological concern, namely that many people do not hold issue attitudes at all. Satisficing is a common type of behaviour on surveys. Those without an attitude may often choose midpoint positions, and may choose those consistently over time. If we compared those who truly hold an attitude, would the findings be the same? I am not sure. At the very least, it would be important to take political interest, engagement, levels of education or cognitive abilities into account in the analysis.**”

People who do not hold attitudes at all: The LISS (Study 2) allows participants without an attitude to select the response options “Don’t know” or “No answer”. We coded those as missing (p. 23 and also in Table 3 in the manuscript). Thus, participants who explicitly said they did not have an attitude about an issue could not have been responsible for the results.

Additional control variables: We conducted additional analyses including the variables you list above as control variables. Specifically, Study 1 additionally included gender, age, education and voting behavior in the last presidential elections; Study 2 additionally included gender, age,

education, voting behavior in the last parliamentary elections and political interest. Unfortunately, neither of the datasets measured cognitive ability. The coding of the variables is explained in the footnotes to the respective results tables (see Tables 9S-12S, pp. 34-42 in the SOM). Adding these control variables did not substantively change the results.

R3 writes: “- Differential dropout rates: Do the authors have information on whether those with extreme views are more likely to drop out in between waves? This is another alternative explanation.”

To test whether individuals who scored on the extreme ends (vs. in the middle) at wave one were more or less likely to drop out, we conducted logistic regression models with attrition (1 = dropped out, 0 = did not drop out) as the dependent variable and a linear and a quadratic terms of attitudes at the first wave as predictors. We conducted separate models for each attitude item. In Study 1, none of 9 models (8 policy issues + political ideology) showed a significant quadratic effect. In Study 2, we found a significant quadratic term for 3 out of 11 models (10 policy issues + political ideology; indicating the extremes were less likely to drop out than moderates). Given that this effect emerged only in 3 (out of overall 20) models and the respective p-values were relatively close to .05 (see Tables 13S-14S and Figure 7S, pp. 43-45 in the SOM), we conclude that the potentially differential attrition rate is unlikely to affect our conclusions.

R3 writes: “5. Smaller points:

- Graphs should always include confidence intervals, yet none of the graphs with quadratic effects do.”

We added confidence bands to our graphs, Figures 1, 3, 4, and 6 in the manuscript.

R3 writes: “- I am not convinced by imposing a quadratic association on all regressions. I would prefer to (also) see regressions with indicator variables for different levels. Imposing quadratic terms can lead to unusual consequences for predicted levels.

As we note in our response to Reviewer 2, Point 7, we conducted an additional set of analyses in which we treated both the time variable and the attitude variables as categorical. The time variable included three categories representing the three time points. For categorizing the attitude variables, we followed the strategy used in Figure 2 (Study 1) and Figure 4 (Study 2): attitudes measured with multi-item scales were split into three categories: bottom 10%, moderate (middle 20%-90%) and top 10%. The model included time, initial attitudes and the interaction terms among them. The significance of the interaction terms indicate that the time effects are moderated by the initial attitudes. The coefficients are presented in Table 9S and the simple slope analysis (i.e., the effect of time for individuals who score low, moderately or high at baseline) results are presented in Table 10S (pp. 21-24 in the SOM).

The results support the conclusions from the analyses using quadratic terms. Specifically, one can see that for individuals with extremely low (vs. high) original attitudes, the effect of time is significant and positive (vs. negative), while for individuals with moderate original attitudes, the effect of time is either nonsignificant or is substantially smaller. We hope these additional analyses are helpful.

We thank Reviewer 3 for the very helpful comments.

3rd Sep 23

Dear Professor Klein,

Thank you for your patience during the peer-review process. Your manuscript titled "The evolution of political extremists: In policy and ideology, extreme attitude holders tend to converge towards the middle over time" has now been seen by 3 reviewers, and I include their comments at the end of this message. It is the same reviewers as before. Two of them have no remaining requests, but the third reviewer raised some important points. Before we make a final decision on publishing your paper we would like to consider your responses to these concerns.

We therefore invite you to revise and resubmit your manuscript, along with a point-by-point response to the reviewers. Please highlight all changes in the manuscript text file.

Addressing the referees' concerns will require some textual revisions (especially with an eye on improving clarity), but also an additional support analysis.

Furthermore, at this stage it is important that you align your manuscript with our format requirements, which are summarized on the following checklist:

<https://www.nature.com/documents/commspsychol-style-formatting-checklist-article-rr.pdf>>Communications Psychology formatting checklist

and also in our style and formatting guide Communications Psychology formatting guide .

Please use the following link to submit your revised manuscript, point-by-point response to the referees' comments (which should be in a separate document to any cover letter) and the completed checklist:

[link redacted]

We hope to receive your revised paper within 8 weeks; we would appreciate it if you could keep us informed about an estimated timescale for resubmission, to facilitate our planning.

Please do not hesitate to contact me if you have any questions or would like to discuss these revisions further. We look forward to seeing the revised manuscript and thank you for the opportunity to review your work.

Best regards,

Yafeng Pan

Yafeng Pan, PhD
Editorial Board Member
Communications Psychology
orcid.org/0000-0002-5633-8313

EDITORIAL POLICIES AND FORMATTING

Editorial Policy: <https://www.nature.com/documents/nr-editorial-policy-checklist.pdf> Policy requirements (Download the link to your computer as a PDF.)

* **CODE AVAILABILITY:** All Communications Psychology manuscripts must include a section titled "Code Availability" at the end of the methods section. In the event of publication, we require that the custom analysis code supporting your conclusions is made available in a publicly accessible repository; at publication, we ask you to choose a repository that provides a DOI for the code; the link to the repository and the DOI will need to be included in the Code Availability statement. Publication as Supplementary Information will not suffice. We ask you to prepare code at this stage, to avoid delays later on in the process.

* **DATA AVAILABILITY:**
All Communications Psychology manuscripts must include a section titled "Data Availability" at the end of the Methods section or main text (if no Methods). More information on this policy, is available at <http://www.nature.com/authors/policies/data/data-availability-statements-data-citations.pdf>.

At a minimum the Data availability statement must explain how the data can be obtained and whether there are any restrictions on data sharing. Communications Psychology strongly endorses open sharing of data. If you do make your data openly available, please include in the statement:

- Unique identifiers (such as DOIs and hyperlinks for datasets in public repositories)
- Accession codes where appropriate
- If applicable, a statement regarding data available with restrictions

- If a dataset has a Digital Object Identifier (DOI) as its unique identifier, we strongly encourage including this in the Reference list and citing the dataset in the Data Availability Statement.

We recommend submitting the data to discipline-specific, community-recognized repositories, where possible and a list of recommended repositories is provided at <http://www.nature.com/sdata/policies/repositories>.

If a community resource is unavailable, data can be submitted to generalist repositories such as <https://figshare.com/> or <http://datadryad.org/> Dryad Digital Repository. Please provide a unique identifier for the data (for example a DOI or a permanent URL) in the data availability statement, if possible. If the repository does not provide identifiers, we encourage authors to supply the search terms that will return the data. For data that have been obtained from publicly available sources, please provide a URL and the specific data product name in the data availability statement. Data with a DOI should be further cited in the methods reference section.

REVIEWERS' COMMENTS:

Reviewer #1 (Remarks to the Author):

I have re-read the paper and the responses to the first set of reviews. I am satisfied with the changes made, and therefore recommend that this paper be accepted for publication in Communications Psychology.

Reviewer #2 (Remarks to the Author):

Recommend: Minor Revision

Response to authors

The authors have addressed most of my concerns and the manuscript has improved a lot. I still have a few remaining questions.

1. The authors added new mechanism explanations for why extremists would turn moderate. I appreciate most of them. But I don't think the temporary situations can explain what is in the data. If individuals' extreme attitudes are changed by events in life, what we should see is extremity spikes, not decreasing trend of extremity.
2. A primary concern was the possibility of regression to the mean. The authors did a good job of improving upon my suggestion and disqualifying measurement error. I think part of the lack of clarity in my response was due to my mixing up a few different ways of disqualifying regression to the mean. The one that the authors pursued and improved upon was to compare how different GSS items do or do not correlate in their extremeness between timesteps. The other comparison I tried to name was to perform the same kind of comparison of potentially correlated variables, between

people who went moderate->extreme and those that went extreme->moderate. But I think the authors' approach is better.

3. Another concern I had was the definition of extremist in the paper. The author explains the reason of analyzing single items and general ideology. The explanation makes sense for the analysis.

However, if those dimensions are not correlated and the questions are asked on a scale of 4 or 5, isn't it too strong a claim to call them extreme attitude holders? To make that claim, it seems more appropriate to do the analysis on general ideology first and then analyze single items to elaborate.

4. Concerns about trend variable and quadratic time term. Now I see what the author means by their trend variable. My suggestion previously was to include a time fixed effect γt . The trend variable in the authors' second set of analyses is b_1X , which controls for any linear trend in the time series. The authors then also included a quadratic trend b_2X^2 for nonlinear effects. I still want to mention that quadratic trend is very rare because it assumes the shape of time effects. Then the author's analysis of the direction of extreme individuals' attitudes is on the interaction with this particular quadratic time effect. To lose these assumptions, in my opinion, it's better to include a fixed time effect and an interaction with the fixed time effect instead. Nevertheless, I agree that the additional analysis result is consistent with the original regression result.

Reviewer #3 (Remarks to the Author):

I am happy that the reviewers took my questions seriously and provided clear answers. I now support publication.

Manuscript #COMMSPSYCHOL-23-0104A.

Responses to Reviewers' Comments

Reviewer #1 (Remarks to the Author):

R1 writes: **“I have re-read the paper and the responses to the first set of reviews. I am satisfied with the changes made, and therefore recommend that this paper be accepted for publication in Communications Psychology.”**

We thank R1 for the helpful feedback in the review process.

Reviewer #2 (Remarks to the Author):

R2 writes: **“Recommend: Minor Revision**

Response to authors

The authors have addressed most of my concerns and the manuscript has improved a lot. I still have a few remaining questions.

1. The authors added new mechanism explanations for why extremists would turn moderate. I appreciate most of them. But I don't think the temporary situations can explain what is in the data. If individuals' extreme attitudes are changed by events in life, what we should see is extremity spikes, not decreasing trend of extremity.”

Several reasons lead us to believe that our data as a whole would not exhibit extremity spikes as a result of the effect of life events on policy attitudes.

a. Any extremity spikes in attitudes will happen right after the relevant life events. Because we identify extreme attitude holders at the onset of our dataset (i.e. the first wave of the longitudinal surveys in Studies 1-2), these extreme attitude holders will have already experienced the relevant life event(s) that presumably caused their attitudes to become extreme. Our analysis therefore captures attitude change that occurs after these life events had happened.

For example, if new parents indeed become “tough on crime” (as existing research suggests, see p. 4 in the manuscript), then we will identify them as extreme attitude holders in the first wave of our data. We then are able to test whether such parents become more extreme or more moderate over subsequent waves (as their children grow up). Our participants certainly could have experienced further life events during the duration of the study. However, our analyses model change depending on individuals' initial scores, not depending on whether they experienced any significant life event later on.

Put differently, we group participants based on extreme vs. moderate initial attitudes and in doing so we observe average attitude trajectories of these groups, which can hide potential spikes experienced by specific individuals.

b. Life events are but one mechanism among several (see pp. 4-5 in the manuscript for possibilities). These other mechanisms imply different trajectories for the evolution of policy attitudes. In the data we likely see the cumulative effects of these various mechanisms, and so it is difficult to isolate the trajectory of the evolution of extreme attitude holders associated with life events.

This logic leads us to believe that life events can be a viable mechanism for the moderation of some extreme attitude holders, and therefore deserves mention in the paper. Having said this, and given that we provide additional ideas for other mechanisms, if there is a strong feeling that life events are not a reasonable mechanism, we are willing to remove this mechanism from the text.

R2 writes: “2. A primary concern was the possibility of regression to the mean. The authors did a good job of improving upon my suggestion and disqualifying measurement error. I think part of the lack of clarity in my response was due to my mixing up a few different ways of disqualifying regression to the mean. The one that the authors pursued and improved upon was to compare how different GSS items do or do not correlate in their extremeness between timesteps. The other comparison I tried to name was to perform the same kind of comparison of potentially correlated variables, between people who went moderate->extreme and those that went extreme->moderate. But I think the authors’ approach is better.”

We are glad that the issue is cleared up and are grateful for the helpful suggestions.

R2 writes: “3. Another concern I had was the definition of extremist in the paper. The author explains the reason of analyzing single items and general ideology. The explanation makes sense for the analysis. However, if those dimensions are not correlated and the questions are asked on a scale of 4 or 5, isn’t it too strong a claim to call them extreme attitude holders? To make that claim, it seems more appropriate to do the analysis on general ideology first and then analyze single items to elaborate.”

We understood this comment as stating that it is more appropriate to first provide the analysis of general ideology and then the analysis of attitudes related to specific policy issues. We chose to follow the opposite order mainly because we had pre-registered the analysis of specific policy issues, whereas the analysis of general ideology was not pre-registered (it had not occurred to us to do it until after the pre-registration).

Given that both analyses of specific policy attitudes and of general ideology ultimately lead to the same general conclusion, we do not believe that the order of presentation affects our theory.

We do believe that presenting the pre-registered analysis before the exploratory analysis is more sound as an empirical approach, and so we left the order of presentation as it is. We are willing to change the order of presentation if there are strong feelings about this issue.

Finally, in terms of identifying extreme vs. moderate attitude holders, we focus on the distribution of attitudes for each item. Specifically, we classify the top or bottom 10% of the distribution of attitude holders as extreme and the rest (the 20% - 80% middle) as moderate. This helps us avoid relying too much on bounded 5-point or 4-point scales, because our classification of extreme vs. moderate is done by comparing participants to each other. In addition, we had added further analyses wherein attitude values were standardized within each wave. In this case, each individual value reflects a person's position relative to others in each respective wave (and not just their response on a 4-5 scale point). These analyses also showed that individuals who held more extreme attitudes (compared to others) initially, became less extreme (compared to others) in the follow-up (see Tables 10S and 11S in the SOM, pp. 31-33).

We believe this adequately captures the definition of “extreme” as residing on the edges of the distribution.

R2 writes: “4. Concerns about trend variable and quadratic time term. Now I see what the author means by their trend variable. My suggestion previously was to include a time fixed effect γt . The trend variable in the authors’ second set of analyses is b1X, which controls for any linear trend in the time series. The authors then also included a quadratic trend b2X2 for nonlinear effects. I still want to mention that quadratic trend is very rare because it assumes the shape of time effects. Then the author’s analysis of the direction of extreme individuals’ attitudes is on the interaction with this particular quadratic time effect. To lose these assumptions, in my opinion, it’s better to include a fixed time effect and an interaction with the fixed time effect instead. Nevertheless, I agree that the additional analysis result is consistent with the original regression result.”

We provide the analysis requested by the reviewer in the SOM, Tables 9S (pp. 21-23 in the SOM).

We are grateful for the helpful suggestions.

Reviewer #3 (Remarks to the Author):

R3 writes: “I am happy that the reviewers took my questions seriously and provided clear answers. I now support publication.”

We thank R3 for the helpful feedback in the review process.

9th Oct 23

Dear Professor Klein,

Your revised manuscript titled "The evolution of political extremists: In policy and ideology, extreme attitude holders tend to converge towards the middle over time" has been evaluated by Dr Yafeng Pan and myself. I am delighted to say that we are happy, in principle, to publish a suitably revised version in Communications Psychology under the open access CC BY license (Creative Commons Attribution v4.0 International License).

We therefore invite you to revise your paper one last time to address the remaining concerns and editorial requests. At the same time we ask that you edit your manuscript to comply with our format requirements and to maximise the accessibility and therefore the impact of your work.

As detailed in the attached checklist, you will need to undertake quite a few presentational changes, including a restructuring of the text. You will also need to prepare the Code, as well as - at least - a minimal dataset for public deposition. These depositions must be linked to in the mandatory Code Availability statement and Data availability statement.

Because it leads very often to delays, we ask you to pay very close attention to the specifications regarding Code/Data sharing, "End Matter" statements (see Requests table), language and statistics. Likewise, your Reporting Summary and Editorial Policy Checklist will be closely inspected on resubmission, so please revise these carefully, using the information that is displayed in the active PDFs. If you have questions about any of these requests or documents, please reach out to us.

Please note that it may still be possible for your paper to be published before the end of 2023, but in order to do this we will need you to address these points as quickly as possible so that we can move forward with your paper.

EDITORIAL REQUESTS:

SUBMISSION INFORMATION:

OPEN ACCESS:

Communications Psychology is a fully open access journal. Articles are made freely accessible on

publication under a [CC BY license](http://creativecommons.org/licenses/by/4.0) (Creative Commons Attribution 4.0 International License). This license allows maximum dissemination and re-use of open access materials and is preferred by many research funding bodies.

For further information about article processing charges, open access funding, and advice and support from Nature Research, please visit <https://www.nature.com/commspsychol/article-processing-charges>

At acceptance, you will be provided with instructions for completing this CC BY license on behalf of all authors. This grants us the necessary permissions to publish your paper. Additionally, you will be asked to declare that all required third party permissions have been obtained, and to provide billing information in order to pay the article-processing charge (APC).

* **DATA AVAILABILITY:**

[link redacted]

Best regards,

Marike

Marike Schiffer, PhD
Chief Editor
Communications Psychology